# Distinct retrograde microtubule motor sets drive early and late endosome transport

Giulia Villari[1,2] (iD), Chiara Enrico Bena[2,3,†] (iD), Marco Del Giudice[2,3] (iD), Noemi Gioelli[1,2] (iD), Chiara Sandri[1,2] (iD), Chiara Camillo[1,2] (iD), Alessandra Fiorio Pla[4], Carla Bosia[3,5] (iD) & Guido Serini[1,2,*] (iD)

## Abstract

**Although subcellular positioning of endosomes significantly impacts on their functions, the molecular mechanisms governing the different steady-state distribution of early endosomes (EEs) and late endosomes (LEs)/lysosomes (LYs) in peripheral and perinuclear eukaryotic cell areas, respectively, are still unsolved. We unveil that such differences arise because, while LE retrograde transport depends on the dynein microtubule (MT) motor only, the one of EEs requires the cooperative antagonism of dynein and kinesin-14 KIFC1, a MT minus end-directed motor involved in cancer progression. Mechanistically, the Ser-x-Ile-Pro (SxIP) motif-mediated interaction of the endoplasmic reticulum transmembrane protein stromal interaction molecule 1 (STIM1) with the MT plus end-binding protein 1 (EB1) promotes its association with the p150Glued subunit of the dynein activator complex dynactin and the distinct location of EEs and LEs/LYs. The peripheral distribution of EEs requires their p150Glued-mediated simultaneous engagement with dynein and SxIP motif-containing KIFC1, via HOOK1 and HOOK3 adaptors, respectively. In sum, we provide evidence that distinct minus end-directed MT motor systems drive the differential transport and subcellular distribution of EEs and LEs in mammalian cells.**

**Keywords** endoplasmic reticulum; motor; movement; traffic; vesicles
**Subject Categories** Cell Adhesion, Polarity & Cytoskeleton; Membranes & Trafficking
**The EMBO Journal (2020) 39: e103661**

## Introduction

In eukaryotic cells, the endolysosomal system is central in carrying out fundamental functions, such as plasma membrane (PM) remodeling, ligand-activated receptor signaling, and acquisition of nutrients (Wideman *et al*, 2014). Once internalized from the cell surface, cargos first localize into the early endosomal compartment (Naslavsky & Caplan, 2018). Early endosomes (EEs) are characterized by the presence of the small GTPase Rab5 that, via phosphatidylinositol 3 (PI3) kinase VPS34, elicits the synthesis of PI3-phosphate (PI3P), allowing the recruitment of PI3P-binding Rab5 effectors, such as early endosome antigen 1 (EEA1) on EEs (Galvez *et al*, 2012). Next, endocytosed cargos are either recycled back to the PM or kept in EEs that, moving along microtubules (MTs), from the cell periphery toward the juxtanuclear MT organizing center (MTOC), undergo maturation into Rab7, PI3,5P2, and lysosomal-associated membrane protein 1 (LAMP-1) containing late endosomes (LEs) and then in degrading lysosomes (LYs). As a result, EEs, which are small (60–400 nm) and weakly acidic (pH 6.8–5.9), localize, at steady state, more peripherally than large (250–1,000 nm) and acidic (pH 6.0–4.9) perinuclear LEs/LYs (Huotari & Helenius, 2011).

Several evidences indicate that the positioning of endosomes within the cytoplasm substantially affects their function (Bonifacino & Neefjes, 2017; Neefjes *et al*, 2017). Directed cell motility depends on the ability to recycle specific integrins and growth factor tyrosine kinase receptors with faster or slower kinetics, in response to their endocytic exit rate from more peripheral or perinuclear sorting compartment stations, respectively (Wilson *et al*, 2018). Cross-presentation to cytotoxic T cells of exogenous antigens, endocytosed by dendritic cells, relies on innate immunity signals that control EE movement along MTs and maturation into LEs/LYs (Weimershaus *et al*, 2018). Albeit so far described only in fungi, hitchhiking emerges as a novel strategy by which molecules (such as mRNA and proteins) and organelles (e.g., peroxisomes, endoplasmic reticulum, and lipid droplets) may connect to and exploit EEs to be evenly distributed throughout the cell (Higuchi *et al*, 2014; Salogiannis & Reck-Peterson, 2017). Recently, precursor miRNAs were also discovered to traffic along axons docked on LEs/LYs to reach growth cones and allow steering by guidance cues and the development of neural circuits (Corradi *et al*, 2020). Both in neuronal (Gowrishankar *et al*, 2015) and non-neuronal (Johnson *et al*, 2016) cells, more peripheral LYs are less proteolytic than the majority of

1  Department of Oncology, University of Torino School of Medicine, Candiolo, Italy
2  Candiolo Cancer Institute - Fondazione del Piemonte per l'Oncologia (FPO), Istituto di Ricovero e Cura a Carattere Scientifico (IRCCS), Candiolo, Italy
3  IIGM - Italian Institute for Genomic Medicine, c/o, IRCCS, Candiolo, Italy
4  Department of Life Sciences and Systems Biology, University of Torino, Torino, Italy
5  Department of Applied Science and Technology, Polytechnic of Torino, Torino, Italy
*Corresponding author. Tel: +39 0119933508; E-mail: guido.serini@ircc.it
†Present Address: Sorbonne Université, CNRS, Institut de Biologie Paris-Seine, Laboratoire Jean Perrin (LJP), Paris, France

LYs, more closely localized around the nucleus, because of either a reduced enzymatic amount (Gowrishankar *et al*, 2015) or activation (Johnson *et al*, 2016) of luminal proteases that have low pH optima (Mellman *et al*, 1986). Furthermore, lysosomal positioning controls the activation of mTOR complex 1 (mTORC1) signaling, which in turn influences autophagosome formation (Korolchuk *et al*, 2011; Poüs & Codogno, 2011). However, the molecular mechanisms responsible for the differential steady-state distribution of EEs and LEs/LYs in peripheral and perinuclear areas of eukaryotic cells, respectively, are unknown.

The cytosolic logistics of endolysosomal cargos relies on motor proteins that move toward both peripheral fast-growing MT plus end and perinuclear slow-growing minus end (Bonifacino & Neefjes, 2017; Neefjes *et al*, 2017; Cross & Dodding, 2019). While endosomes are transported along MTs centrifugally (anterograde transport) by several kinesin motors, so far cytoplasmic multiprotein dynein 1 complex (hereafter referred as dynein) has been identified as the only main MT motor in charge of the opposite centripetal movement (retrograde transport) (Bonifacino & Neefjes, 2017; Neefjes *et al*, 2017; Cross & Dodding, 2019). Indeed, specific kinesins drive the anterograde motion of either EEs and LEs/LY, whereas their retrograde transport is thought to depend on the coupling of the same dynein motor to different adaptor proteins (Reck-Peterson *et al*, 2018). To function as a highly processive motor, dynein must undergo conformational transition from an auto-inhibited to an active conformational state (Zhang *et al*, 2017; McKenney, 2018). The multiprotein asymmetric complex dynactin is the main cofactor that releases dynein from its auto-inhibition at MT plus end and activates its minus end-directed retrograde motility (Ketcham & Schroer, 2018; McKenney, 2018). Distinct adaptor proteins containing a long coiled coil (CC) domain, such as bicaudal D cargo adaptor 2 (BICD2) (Hoogenraad & Akhmanova, 2016) as well as the Rab11 family interacting protein 3 (FIP3), HOOK3, and the spindle apparatus coiled coil protein 1 (SPDL1) (McKenney *et al*, 2014), are required to stabilize the formation at the MT plus end of the dynein–dynactin complex, bound to specific cargos, and prompted to drive their retrograde transport (McKenney, 2018; Reck-Peterson *et al*, 2018).

Proteins localized at the MT plus end, acting as scaffolds where dynactin is recruited and thereby contributing to dynein conformational activation, play a key role in the initiation of movement toward the MT minus end (Akhmanova & Steinmetz, 2015; McKenney, 2018). Dynactin is formed by a short both-side capped actin-related protein 1 (ARP1) polymer and a projecting p150Glued side arm, kept together by a shoulder complex (Ketcham & Schroer, 2018; Reck-Peterson *et al*, 2018). Three CC stretches allow p150Glued to extend from the shoulder complex and interact with the MT plus end via its basic and cytoskeleton-associated protein glycine-rich (CAP-Gly) domain (Ketcham & Schroer, 2018; Reck-Peterson *et al*, 2018). The CAP-Gly domain located at the N-terminus of p150Glued interacts with the C-terminal Glu-Glu-Tyr (EEY) motif of tyrosinated α-tubulin and end-binding (EB) proteins, both enriched at the MT plus end (Akhmanova & Steinmetz, 2015; McKenney *et al*, 2016; McKenney, 2018; Rupam & Surrey, 2018). Yet, how interactions among MT plus end proteins, dynactin, dynein, and different endosomal cargos are coordinated and regulated in cells is still under investigation (Olenick *et al*, 2019; Saito *et al*, 2020). Moreover,

differently from fungi (Steinberg, 2014), the molecular machinery that moves EEs toward the nucleus in animal cells is only in part understood (Neefjes *et al*, 2017).

Here, we reveal a new role for the endoplasmic reticulum (ER) transmembrane protein stromal interaction molecule 1 (STIM1), previously shown to bind via its Ser-x-Ile-Pro (SxIP) motif the protein EB1 (Grigoriev *et al*, 2008), in promoting the association of the p150Glued subunit of dynactin to EB1 and its ensuing recruitment to the plus end of MTs in mammalian cells. As a result, STIM1 plays a key role in the regulation of endosomal cargo loading and dynein-dependent transport. We also unveil that, while LEs are transported toward the nucleus by the STIM1-dependent recruitment of the dynactin/dynein complex, the retrograde transport of EEs depends on the antagonistic cooperation between dynein and the minus end-directed kinesin-14 KIFC1, which also binds EB1 through a SxIP motif (Braun *et al*, 2013). The cooperative antagonism between dynein and KIFC1 may thus represent a molecular strategy to differentially regulate MT-based transport and subcellular distribution of specific vesicular components of the endolysosomal system, a general feature that is central for fundamental functions of eukaryotic cells.

## Results

### STIM1 forms a triple protein complex with p150Glued and EB1 to promote dynactin loading at MT plus ends

The dynactin subunit p150Glued plays a key role in the release of the MT plus end protein EB1 auto-inhibition and subsequent engagement of the dynein/dynactin complex at MT plus ends (Hayashi *et al*, 2005; Akhmanova & Steinmetz, 2015; McKenney *et al*, 2016; McKenney, 2018). However, how those molecular dynamics, so far investigated in *in vitro* studies with recombinant proteins or in crystals (Duellberg *et al*, 2014), are regulated in intact living cells is unknown. Considering the ability of the ER protein STIM1 to bind, similarly to p150Glued, EB1 (Grigoriev *et al*, 2008), we verified whether STIM1 silencing by short hairpin (sh) RNA (shSTIM1; Fig EV1A) may affect p150Glued association with the MT plus end protein in human primary endothelial cells (ECs). The lack of STIM1 significantly reduced the amount of p150Glued that interacts with EB1 at steady-state in living ECs (Fig 1A), thus suggesting a potential new role for STIM1 in the ordinated recruitment of proteins at MT plus ends and, potentially, the loading and retrograde transport of cargos (Ayloo *et al*, 2014; Reck-Peterson *et al*, 2018). Yet, STIM1 silencing did not affect the assembly of the dynein/dynactin complex (Fig EV1B).

Next, we wondered whether, other than with EB1 (Grigoriev *et al*, 2008), STIM1 may also associate with p150Glued in ECs. We found that indeed STIM1 co-immunoprecipitates with p150Glued (Fig 1B), but not with dynein light-intermediate chain (LIC), thus suggesting a triple complex formed by the ER protein, EB1, and p150Glued. To directly assess the possible formation of this ternary complex, we performed *in vitro* interaction assays with the corresponding purified proteins. In particular, we generated and purified the wild-type FLAG-tagged C-terminal EB1 portion containing the STIM1-binding EBH domain (Grigoriev *et al*, 2008) and the

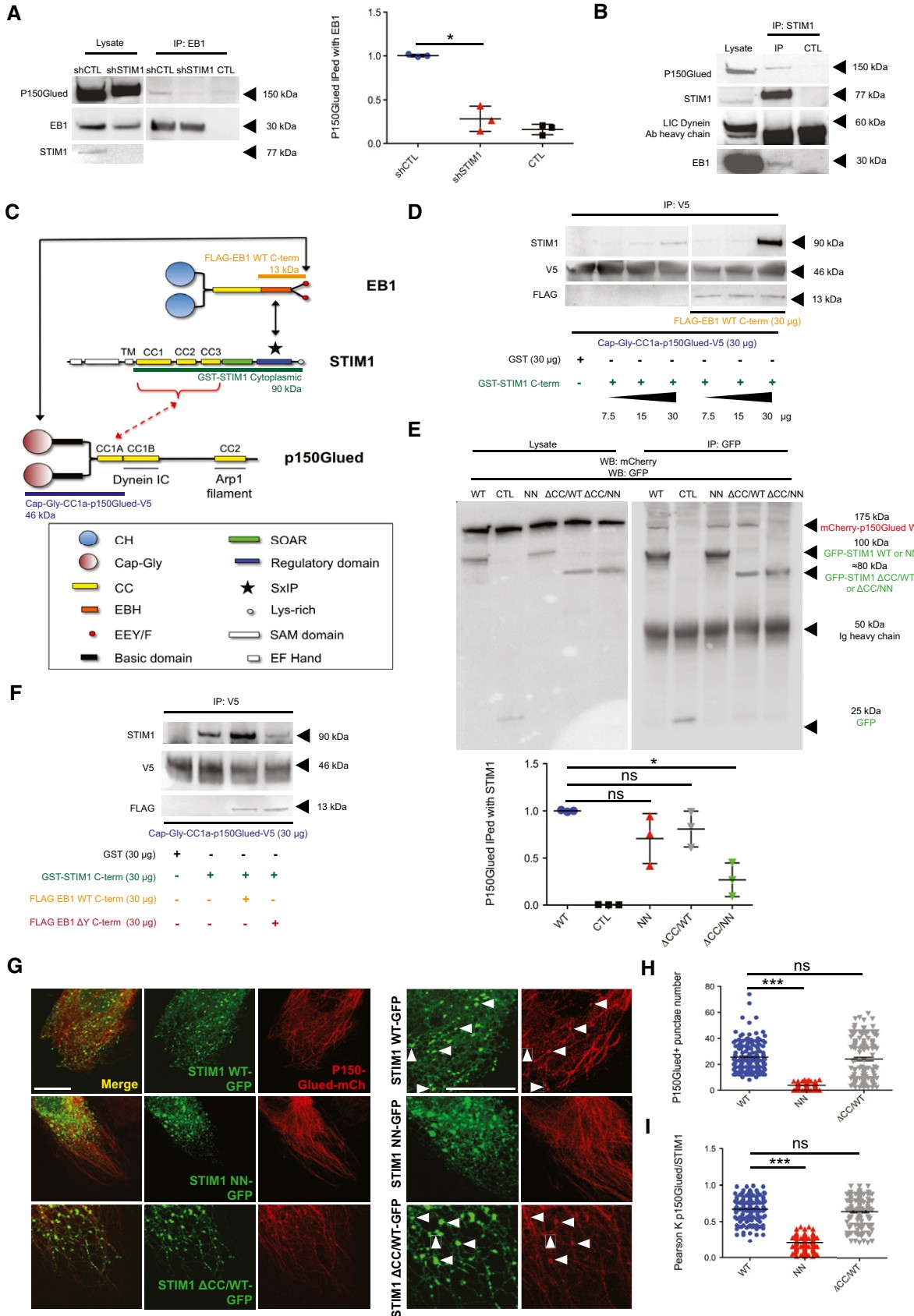

**Figure 1.**

**Figure 1.  STIM1 forms a triple protein complex with p150Glued and EB1 to promote dynactin loading at MT plus ends.**

A   Representative Western blot analysis of the endogenous p150Glued co-immunoprecipitated with EB1 in shCTL or shSTIM1 ECs (left) and its quantification by normalized densitometry (right). Negative control (CTL) was performed incubating cell lysate with protein A- or G-Sepharose and empty mouse IgG. Results are the average $\pm$ SD of three independent assays. shCTL value of each biological replicate was normalized on itself and so shSTIM1 experimental value. Results were analyzed by a parametric two-tailed analysis of variance (ANOVA) with Bonferroni *post hoc* analysis. ANOVA $P \leq 0.001$; Bonferroni for shCTL and shSTIM1 $P \leq 0.05*$.

B   Representative of three Western blot analysis of endogenous p150Glued, light-intermediate chain (LIC) of cytoplasmic dynein 1, and EB1 immunoprecipitated with STIM1 in wild-type ECs. Negative control (CTL) was performed incubating cell lysate with protein A- or G-Sepharose and empty rabbit IgG.

C   Schematic model of the molecular interactions among STIM1, EB1, and p150Glued at the MT plus ends. Black arrows point at already known binding motifs, whereas red parts show interaction domains studied in this manuscript. TM, transmembrane domain. Differentially colored labels with bars correspond to the purified protein fragment with its tag and molecular weight in kDa.

D   Representative Western blot analysis of increasing amount of GST-STIM1 (cytoplasmic domain) pulled down by Cap-Gly-CC1-p150Glued-V5-coated beads, in the absence or presence of FLAG-EB1 C-terminal. Colors label the protein as in (C). Negative control was performed using equal amount of empty GST protein, together with Cap-Gly-CC1-p150Glued.

E   Representative Western blot analysis of mCherry-p150Glued WT co-immunoprecipitated with GFP-STIM1 WT, NN, $\Delta$CC1–3/WT ($\Delta$CC/WT), or $\Delta$CC1–3/NN ($\Delta$CC/NN) in cotransfected HEK 293T cells. Negative control (CTL) was performed incubating cell lysate from HEK 293T cotransfected with an empty GFP vector together with mCherry-p150Glued WT with pre-cleared protein A or G-Sepharose and the rabbit GFP antibody. Below, its quantification by normalized densitometry. Results are the average $\pm$ SD of three independent assays. The value of p150Glued co-immunoprecipitated with GFP-STIM1 WT from each biological replicate was normalized on itself and so those immunoprecipitated with GFP-STIM1 NN, $\Delta$CC/WT or $\Delta$CC/NN. Results were analyzed by a parametric two-tailed analysis of variance (ANOVA) with Bonferroni *post hoc* analysis. ANOVA $P \leq 0.001$; Bonferroni for STIM1 WT and NN $P > 0.05$ not significant (ns), for STIM1 WT and $\Delta$CC/WT $P > 0.05$ not significant (ns) and for STIM1 WT and $\Delta$CC/NN $P \leq 0.05*$.

F   Representative Western blot analysis of GST-STIM1 (cytoplasmic domain) pulled down by Cap-Gly-CC1-p150Glued-V5-coated beads, in the presence of FLAG-EB1 C-terminal WT or $\Delta$Y. Negative control was performed using equal amount of empty GST protein, together with Cap-Gly-CC1-p150Glued.

G   Confocal microscopy analysis of wild-type ECs, transiently transfected with GFP-STIM1 WT or, NN or $\Delta$CC1–3/WT ($\Delta$CC/WT) together with mCherry-p150Glued. Scale bar = 10 μm. Right insets are shown to highlight MT-bound p150Glued$^+$ punctae (arrows), colocalized with STIM1. Scale bar = 5 μm.

H   Average number of the mCherry-p150Glued$^+$ punctae in the same cells as in (G). Counts are the average $\pm$ SEM of three independent experiments for a total of 150 punctae (30 cells). Results were analyzed by a parametric two-tailed analysis of variance (ANOVA) with Bonferroni *post hoc* analysis. ANOVA $P \leq 0.001$; Bonferroni for STIM1 WT and NN $P \leq 0.001***$ and for STIM1 WT and $\Delta$CC/WT $P > 0.05$ not significant (ns).

I    Colocalization analysis of mCherry-p150Glued with GFP-STIM1 WT, NN, or $\Delta$CC1–3/WT ($\Delta$CC/WT), transiently cotransfected in ECs, as in (G). Results are the average $\pm$ SEM of three independent experiments for a total of 150 punctae (30 cells) and analyzed by a parametric two-tailed analysis of variance (ANOVA) with Bonferroni *post hoc* analysis. ANOVA $P \leq 0.001$; Bonferroni for STIM1 WT and NN $P \leq 0.001***$ and for STIM1 WT and $\Delta$CC/WT $P > 0.05$ not significant (ns).

Source data are available online for this figure.

p150Glued-interacting EEY motif (Akhmanova & Steinmetz, 2015) (FLAG-EB1 WT C-term, orange bar in Fig 1C), the GST-tagged cytoplasmic domain of STIM1 (GST-STIM1 cyto, green bar in Fig 1C), and the Cap-Gly domain of p150Glued fused with its first CC (CC1a), V5-tagged (Cap-Gly-CC1a-p150Glued-V5, blue bar in Fig 1C). In the last construct, we included the CC1a domain alone, as we posited it would have been the only one, among the three CC regions of p150Glued (Tripathy *et al*, 2014), available for the binding to STIM1, being the second (CC1b) and third (CC2) domain known to interact with dynein IC and the Arp1 filament of dynactin, respectively (McKenney, 2018; Reck-Peterson *et al*, 2018). Moreover, since the CC1a is known to exist in an inhibited form folded with the second CC1b motif, using the first part only of the whole CC1 of p150Glued would have allowed us to avoid any inactivation of the protein (Wang *et al*, 2014; Saito *et al*, 2020). We confirmed *in vitro* the binding between purified Cap-Gly-CC1a-p150Glued-V5 and GST-STIM1 cyto, with increasing amount of the latter detected to associate with immunoprecipitated Cap-Gly-CC1a-p150Glued-V5 (Fig 1D). Of note, the *in vitro* interaction between p150Glued and STIM1 was clearly stabilized by the addition of purified FLAG-EB1 WT C-term, which is known to bind both the Cap-Gly domain of p150Glued (Akhmanova & Steinmetz, 2015; McKenney *et al*, 2016; McKenney, 2018; Rupam & Surrey, 2018) and the SxIP motif of STIM1 (Grigoriev *et al*, 2008). Hence, the ER protein STIM1 forms a ternary complex together with the MT plus end protein EB1 and the dynactin subunit p150Glued both in living ECs and *in vitro*.

Then, we verified whether, similarly to what observed *in vitro* with purified recombinant proteins (Fig 1D), also in living ECs the interaction between STIM1 and p150 Glued relies on the simultaneous binding of STIM1 to EB1. The SxIP motif mediates the association of STIM1 to the EB homology (EBH) domain of EB1 and the ensuing STIM1 tracking of MT plus ends (Yao *et al*, 2012; Chowdhury *et al*, 2015). To verify the possible involvement of STIM1 SxIP motif in the interaction with p150Glued, we employed a mutant version of STIM1 unable to bind EB1 (Grigoriev *et al*, 2008; Honnappa *et al*, 2009) and dubbed GFP-STIM1 NN (Fig 1C) as the Ile and Pro residues of the SxIP motif were replaced by two Asn (N). Since STIM1 also owns three CC domains (Novello *et al*, 2018), which are known ubiquitous protein–protein interaction domains (Woolfson *et al*, 2012) and key in releasing the auto-inhibition of MT plus end proteins (Hayashi *et al*, 2005), we asked whether STIM1 CC motifs may also mediate its association with p150Glued. Hence, we generated GFP-STIM1 mutants lacking each of the CC domain ($\Delta$CC1, $\Delta$CC2, and $\Delta$CC3; Fig EV1C) or all of them ($\Delta$CC1–3; Fig 1E) in both wild-type (WT) or NN-STIM1 backbones ($\Delta$CC1–3/WT and $\Delta$CC1–3/NN), cotransfected them with mCherry-p150Glued WT in HEK239T cells, and verified their interactions. Neither the deletion of single (Fig EV1C) or all three CC domains ($\Delta$CC1–3/WT; Fig 1E) nor the mutation of the SxIP motif (NN; Fig 1E) affected STIM1 binding to p150Glued, compared to WT STIM1 in living cells. However, the GFP-STIM1 $\Delta$CC1–3/NN mutant, which simultaneously lacks all CC domains and the ability to bind EB1 via the SxIP motif, did not co-immunoprecipitate with p150Glued (Fig 1E). Thus, in agreement with what we observed *in vitro* with purified proteins (Fig 1D), those results in living cells further supported the notion that STIM1 forms a triple protein complex with p150Glued and EB1 via its coiled coil domains and its

SxIP motif. In this context, to further characterize our finding that *in vitro* the addition of EB1 stabilizes and increases the interaction of p150Glued with STIM1, giving rise to the formation of the triple protein interaction (Fig 1D), we generated and purified a mutant FLAG-EB1 C-term construct lacking the tyrosine (Y) residue in the Glu-Glu-Tyr (EEY, red circles in Fig 1C) motif (FLAG-EB1 ΔY C-term), which is responsible for EB1 binding to the Cap-Gly domain of dynactin p150Glued (Komarova *et al*, 2005). Differently from FLAG-EB1 WT C-term (Fig 1D and F), the FLAG-EB1 ΔY C-term mutant was completely unable to stabilize the *in vitro* interaction between GST-STIM1 cyto and Cap-Gly-CC1a-p150Glued-V5 proteins (Fig 1F). Notably, FLAG-EB1 ΔY C-term even substantially decreased the basal amounts of cytoplasmic STIM1 that interact with p150Glued (Fig 1F). In addition to confirming the existence of a triple STIM1-p150Glued-EB1 complex, these data further highlight the cooperative role that the EBH domain and the EEY motif of EB1 play in strengthening the bridging between STIM1 and p150Glued.

To better understand the functional implications of STIM1 complexing with p150Glued and EB1 in living cells, we imaged by fluorescence confocal microscopy ECs cotransfected with different GFP-STIM1 constructs and mCherry-p150Glued WT (Fig 1G). Notably, the overexpression in ECs of the EB-1-interacting constructs STIM1 WT or ΔCC1–3/WT, but not of the EB-1 independent mutant STIM1 NN, elicited the accumulation of p150Glued in bright punctate structures at MT plus ends (Fig 1G and H). Furthermore, quantitative analysis revealed that p150Glued colocalizes with STIM1 WT or ΔCC1–3/WT, but much less with STIM1 NN (Fig 1I). These microscopy data suggest that, although STIM1 NN mutant can still bind p150Glued (Fig 1E), the inability of simultaneously bind EB1, via its SxIP motif, impedes STIM1 to favor the enrichment of p150Glued at EB1 containing MT plus ends and consequently the colocalization of STIM1 and p150Glued at this location. Altogether, those data further highlight the key functional implications of the triple STIM1/EB1/p150Glued complex formation to promote dynactin loading at MT ends.

## STIM1 fosters EB1-loaded late endosome retrograde transport

Altogether, our data substantiated a novel role for STIM1 in promoting p150Glued-containing dynactin complex loading at EB1[+] MT plus ends, thereby suggesting a possible role of STIM1 in the regulation of cargo retrograde transport. The dynein/dynactin complex is the key machinery regulating LE retrograde movement (Ayloo *et al*, 2014; Reck-Peterson *et al*, 2018) and its disruption results in dramatic changes in LE positioning (Yao *et al*, 2012; Chowdhury *et al*, 2015). To elucidate the function that STIM1 may play in the dynein/dynactin-dependent centripetal transport of LEs, we transiently silenced it by short interfering RNA (siRNA) in ECs (siSTIM1; Fig EV1D). Fluorescence confocal microscopy showed that, compared to control oligofected cells (siCTL), siSTIM1 ECs display a substantially different localization of lysosomal-associated membrane protein 1 (LAMP-1)-labeled LEs (Fig 2A). In agreement with the notion that size and number of endosomes depend on their motor-driven positioning in the cell (Aoyama *et al*, 2017; Bonifacino & Neefjes, 2017), computer-assisted automated measurements showed that in siCTL ECs LAMP-1[+] LEs reach typical large sizes, which did not appear in siSTIM1 cells (Fig 2B). Moreover, cumulative distribution function analysis (Kolmogorov–Smirnov test)

revealed that the tail of the distribution of siCTL LEs size is significantly longer (*P*-value ≤ 0.001), than that of siSTIM1 LEs, supporting a decrease of those vesicle size in siSTIM1, compared to siCTL ECs (Fig 2B, inset). In addition, in siSTIM1 ECs the same LEs were increased in number (Fig 2C) and more dispersed throughout the cytoplasm (Fig 2D), losing their characteristic perinuclear localization, observed instead in siCTL ECs (Fig 2A and D). Thus, STIM1, which interacts with and promotes the formation of a complex between EB1 and the dynein cofactor dynactin, also fosters dynein-driven retrograde movement of LEs along MTs.

In addition to the interaction with MT plus end-associated proteins (Grigoriev *et al*, 2008) (this manuscript), STIM1 can sense, via its intraluminal EF hand motifs, the drop of free calcium (Ca[2+]) in ER stores. This event results in STIM1 detachment from MTs, contact and activation of the channel Orai1 at the PM, which in turn lets extracellular Ca[2+] to refill the ER, in a mechanism known as store-operated calcium entry (SOCE) (Yuan *et al*, 2009; Vaca, 2010; Chang *et al*, 2018). To understand whether and how the ability to interact with EB1 and to regulate Ca[2+] homeostasis may influence the control that STIM1 exerts on the retrograde motion of LEs along MTs, we performed rescue experiments in which shSTIM1 ECs were transfected with GFP alone, for control purposes, or silencing-resistant GFP-tagged wild-type (WT) or point mutant STIM1 constructs. In particular, we employed STIM1 NN, which is unable to bind the MT plus end protein EB1 (Fig 1E), and STIM1-AA, in which the two Leu-Gln residues of the STIM1 Orai1-activating region (SOAR) domain, essential for Orai1 binding and activation, were mutated into two Alanines (AA) (Yuan *et al*, 2009; Vaca, 2010; Chang *et al*, 2018). Fluorescence confocal microscopy (Fig 2E) and automated quantitative analysis (Fig 2F–H) revealed that, when compared to control (shCTL), shSTIM1 ECs display more peripheral (Fig 2E and F, upper panel), smaller (Fig 2E and G), and more numerous (Fig 2E and H) LAMP-1[+] LEs. In addition, the overexpression of GFP-STIM1 WT in the same cells was able to rescue those phenotypes, as in controls. Moreover, GFP-STIM1 AA, but not GFP-STIM1 NN, restored perinuclear LE subcellular positioning (Fig 2E and F, lower panel), size (Fig 2E and G), and number (Fig 2E and H). Those data suggest that, although STIM1 NN was still able to bind p150Glued via its CC (Fig 1E), their interaction is not sufficient to promote perinuclear LE localization, observed in STIM1 WT rescued cells, further highlighting the role of the triple bridging complex in the regulation of LE minus end-directed motion. Moreover, these observations further imply that the function of STIM1 in promoting LE retrograde transport clearly relies on its interaction with MT plus end EB proteins, but not on its binding to Orai1. To further study the relationship between STIM1 role in LE positioning and the one in the Orai1-binding dependent regulation of ER Ca[2+] stores (Yuan *et al*, 2009; Vaca, 2010; Chang *et al*, 2018), we treated ECs with the sarco-ER Ca[2+] ATPase (SERCA) inhibitor Thapsigargin (TG), which reduces those stores and induces STIM1 translocation to the PM (Fig EV1E, left). As shown by confocal fluorescence microscopy (Fig EV1E, right) and automated quantitative analysis (Fig EV1F), TG treatment increased LE distance to the nucleus compared to the untreated (UT) counterpart, thus mimicking in full the phenotype observed in siSTIM1 cells. Those data further highlight a potential dual role for STIM1 in the modulation of LE transport, when bound to the MT plus ends, and one in the replenishment of the ER Ca[2+] stores, if interacting with Orai1.

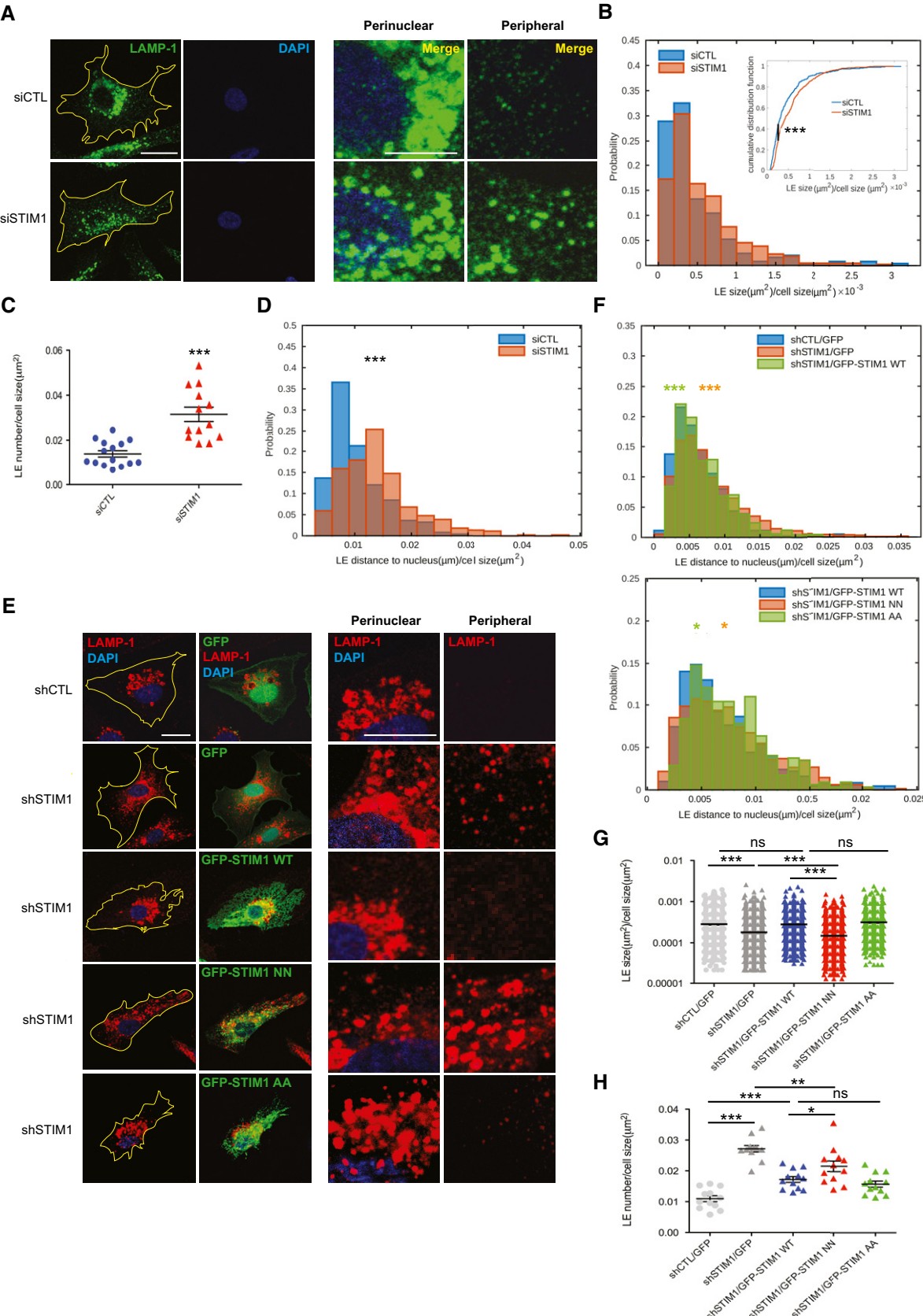

**Figure 2.**

**Figure 2.  STIM1 fosters EB1-dependent late endosome retrograde transport.**

A  Confocal microscopy images of ECs silenced with a control siRNA (siCTL) or one targeting STIM1 (siSTIM1) and stained for endogenous LAMP-1 (in green) to visualize LEs and DAPI (in blue) to highlight the nucleus. The yellow line is drawn to define cell periphery. Scale bar = 20 μm. On the right, inset panels to highlight respective perinuclear and peripheral area of the cell. Scale bar = 5 μm.

B  Distribution of size, normalized on cell size, quantified by image (as in A) segmentation (see Materials and Methods, Confocal microscopy and early/late quantification) of LAMP-1⁺ endosomes. Results are from three independent experiments for a total of 249 endosomes in 15 siCTL cells (17 ± 2 endosomes per cell) and 441 endosomes in 13 siSTIM1 cells (34 ± 3 endosomes per cell). In the right inset, the cumulative distribution used for the Kolmogorov–Smirnov test, $P \leq 0.001$ ***, is shown. The *P*-value refers to the rejection of the null hypothesis in favor of the alternative hypothesis that siCTL LE distribution has longer tail than siSTIM1 LE one.

C  Average number, normalized on cell size, quantified by image (as in A) segmentation (see Materials and Methods, Confocal microscopy and early/late quantification) of LAMP-1⁺ endosomes. Results are the average ± SEM of three independent experiments for a total of 249 late endosomes in 15 siCTL cells (17 ± 2 endosomes per cell) and 441 late endosomes in 13 siSTIM1 cells (34 ± 3 endosomes per cell) and analyzed by a two-tailed heteroscedastic Student's *t*-test, $P \leq 0.001$***.

D  Distribution of distance to nucleus, normalized on cell size, quantified by image (as in A) segmentation (see Materials and Methods, Confocal microscopy and early/late quantification) of LAMP-1⁺ endosomes. Results are from three independent experiments for a total of 249 late endosomes in 15 siCTL cells (17 ± 2 endosomes per cell) and 441 late endosomes in 13 siSTIM1 cells (34 ± 3 endosomes per cell) and analyzed by a two-tailed heteroscedastic Student's *t*-test, $P \leq 0.001$ ***.

E  Confocal microscopy images of ECs silenced with a control shRNA (shCTL) or one targeting STIM1 (shSTIM1), rescued for GFP expression alone (shCTL/GFP and shSTIM1/GFP) or for GFP-STIM1 WT (shSTIM1/GFP-STIM1 WT), GFP-STIM1 NN (shSTIM1/GFP-STIM1 NN), or GFP-STIM1 AA (shSTIM1/GFP-STIM1 AA) and stained for endogenous LAMP-1 (in red) to visualize LEs and DAPI (in blue) to highlight the nucleus. The yellow line is drawn to define cell periphery. Scale bar = 20 μm. On the right, inset panels to highlight respective perinuclear and peripheral area of the cell. Scale bar = 5 μm.

F  Upper panel, Distribution of distance to nucleus, normalized on cell size, quantified by image (as in E) segmentation (see Materials and Methods, Confocal microscopy and early/late quantification) of LAMP-1⁺ endosomes. Results are from of two independent experiments for a total of 436 late endosomes in 13 shCTL/GFP cells (34 ± 6 endosomes per cell), 1,452 late endosomes in 12 shSTIM1/GFP cells (121 ± 13 endosomes per cell), and 498 late endosomes in 10 shSTIM1/GFP-STIM1 WT cells (50 ± 5 endosomes per cell) and analyzed by a parametric two-tailed analysis of variance (ANOVA) with Bonferroni *post hoc* analysis. ANOVA $P \leq 0.01$; Bonferroni for shCTL/GFP and shSTIM1/GFP $P \leq 0.001$*** (orange) and for shSTIM1/GFP and shSTIM1/GFP-STIM1 WT $P \leq 0.001$*** (green). Lower panel, Distribution of distance to nucleus, normalized on cell size, quantified by image (as in E) segmentation (see Materials and Methods, Confocal microscopy and early/late quantification) of LAMP-1⁺ endosomes. Results are from of two independent experiments for a total of 498 late endosomes in 10 shSTIM1/GFP-STIM1 WT cells (50 ± 5 endosomes per cell), 1,057 late endosomes in 11 shSTIM1/GFP-STIM1 NN cells (96 ± 12 endosomes per cell), and 346 late endosomes in 10 shSTIM1/GFP-STIM1 AA cells (35 ± 2 endosomes per cell) and analyzed by a parametric two-tailed analysis of variance (ANOVA) with Bonferroni *post hoc* analysis. ANOVA $P \leq 0.01$; Bonferroni for shSTIM1/GFP-STIM1 WT and shSTIM1/GFP-STIM1 NN $P \leq 0.05$* (orange) and for shSTIM1/GFP-STIM1 NN and shSTIM1/GFP-STIM1 AA $P \leq 0.05$* (green).

G  Average size, normalized on cell size, quantified by image (as in E) segmentation (see Materials and Methods, Confocal microscopy and early/late quantification) of LAMP-1⁺ endosomes. Results are the average ± SEM of two independent experiments for a total of 436 late endosomes in 13 shCTL/GFP cells (34 ± 6 endosomes per cell), 1,452 late endosomes in 12 shSTIM1/GFP cells (121 ± 13 endosomes per cell), 615 late endosomes in 12 shSTIM1/GFP-STIM1 WT cells (51 ± 5 endosomes per cell), 1,278 late endosomes in 12 shSTIM1/GFP-STIM1 NN cells (106 ± 15 endosomes per cell), and 506 late endosomes in 12 shSTIM1/GFP-STIM1 AA cells (42 ± 7 endosomes per cell) and analyzed by a parametric two-tailed analysis of variance (ANOVA) with Bonferroni *post hoc* analysis. ANOVA $P \leq 0.001$; Bonferroni for shCTL/GFP and shSTIM1/GFP $P \leq 0.001$***, for shCTL/GFP and shSTIM1/GFP-STIM1 WT $P > 0.05$ not significant (ns), for shSTIM1/GFP-STIM WT and shSTIM1/GFP-STIM1 NN $P \leq 0.001$***, for shSTIM1/GFP and shSTIM1/GFP-STIM1 NN $P \leq 0.001$*** and for shSTIM1/GFP-STIM WT and shSTIM1/GFP-STIM1 AA $P > 0.05$ not significant (ns).

H  Average number, normalized on cell size, quantified by image (as in E) segmentation (see Materials and Methods, Confocal microscopy and early/late quantification) of LAMP-1⁺ endosomes. Results are the average ± SEM of two independent experiments for a total of 436 late endosomes in 13 shCTL/GFP cells (34 ± 6 endosomes per cell), 1,452 late endosomes in 12 shSTIM1/GFP cells (121 ± 13 endosomes per cell), 615 late endosomes in 12 shSTIM1/GFP-STIM1 WT cells (51 ± 5 endosomes per cell), 1,278 late endosomes in 12 shSTIM1/GFP-STIM1 NN cells (106 ± 15 endosomes per cell), and 506 late endosomes in 12 shSTIM1/GFP-STIM1 AA cells (42 ± 7 endosomes per cell) and analyzed by a parametric two-tailed analysis of variance (ANOVA) with Bonferroni *post hoc* analysis. ANOVA $P \leq 0.001$; Bonferroni for shCTL/GFP and shSTIM1/GFP $P \leq 0.001$***, for shCTL/GFP and shSTIM1/GFP-STIM1 WT $P \leq 0.001$***, for shSTIM1/GFP-STIM WT and shSTIM1/GFP-STIM1 NN $P \leq 0.05$*, for shSTIM1/GFP and shSTIM1/GFP-STIM1 NN $P \leq 0.01$** and for shSTIM1/GFP-STIM WT and shSTIM1/GFP-STIM1 AA $P > 0.05$ not significant (ns).

Source data are available online for this figure.

## STIM1 and dynein oppositely regulate early and late endosome retrograde transport

Next, we investigated whether the role of STIM1 in the control of EB1/dynactin interaction and dynein-dependent transport may impact on EE positioning in ECs. Surprisingly, confocal microscopy revealed that, compared to LEs (Fig 2), both Rab5⁺ and EEA-1⁺ EEs display a reversed phenotype in siSTIM1 ECs (Fig 3A, left panels), being larger, very clustered in the perinuclear area and strongly reduced in number (Fig 3A, right panels). Therefore, we automatically quantified, as we did for LEs, size, number, and distance to the nucleus of EEA-1⁺ EE endosomes. Confocal microscopy analysis showed that, when compared to siCTL, EEs in siSTIM1 ECs are persistently larger (Fig 3B) and decreased in number (Fig 3C). Additionally, time-lapse confocal microscopy coupled to computer-assisted automated detection and tracking of GFP-Rab5⁺ EE endosomes in living siCTL and siSTIM1 ECs (Movies EV1 and EV2), indicated that those vesicles aggregate, overtime, around the nuclei of STIM1 silenced cells (Fig 3D). To understand whether the control of EE retrograde transport may rely on the SxIP-mediated interaction of STIM1 with EB1, we performed rescue experiments (Fig 4A, left panels) and, through automatic quantitative analysis, found that indeed GFP-STIM1 WT overexpression in shSTIM1 ECs (Fig 4B, upper panel) was able to rescue the decreased distance to the nucleus of EEs, observed in those cells. On the other hand, GFP-STIM1 NN (Fig 4B, lower panel) did not modify the abnormal EE perinuclear positioning, observed in STIM1 silenced cells. Moreover, STIM1 WT, but not its NN mutant, was also able to rescue the large size (Fig 4A, right panels, and C) and reduced number (Fig 4A, right panels, and D) of the same vesicles observed in shSTIM1 ECs. Altogether, these data indicate that STIM1 also regulates the retrograde transport of EEs, that, as observed for LEs (Fig 2A–D), depends on its interaction with EB1 at MT plus ends, yet with an entirely reverse effect in terms of motion direction and subcellular localization. Hence, STIM1 appears to play an opposite function in the regulation of LE or EE movement along MTs.

Differently from LEs, the machinery that in mammalian cells controls EE retrograde movement along MTs is so far largely unknown (Li *et al*, 2016). Since we found that STIM1, via its ability to bind EB1 and stabilizing its interaction with p150Glued subunit of dynactin, promotes the retrograde transport of LEs along MTs (Figs 1 and 2), we wondered whether dynein may also participate with STIM1 to control the subcellular distribution of EEs in ECs. Before doing that, to assure that STIM1 role in the regulation of motor function would not affect EE-to-LE maturation, we measured and did not observe any impact of STIM1 silencing on the amount

of EEA1 and LAMP colocalization in endothelial endosomes (Fig EV2A). Next, we treated cells with Ciliobrevin D (CilioD), a specific cell permeable inhibitor of the ATPase activity of dynein (Yao *et al*, 2012; Chowdhury *et al*, 2015), and observed its effect on both EEA1$^+$ EEs and LAMP-1-1$^+$ LEs positioning (Fig 4E, left panels). Notably, CilioD inhibition of dynein phenocopied the effect of STIM1 silencing (Figs 2A and D, and 3A and D), diminishing the distance to the nucleus of EEs (Fig 4E, right panels and F), and, as expected, increasing the one of LEs in ECs (Fig 4E, right panels and G). CilioD similarly affected the positioning of EEs and LEs also in

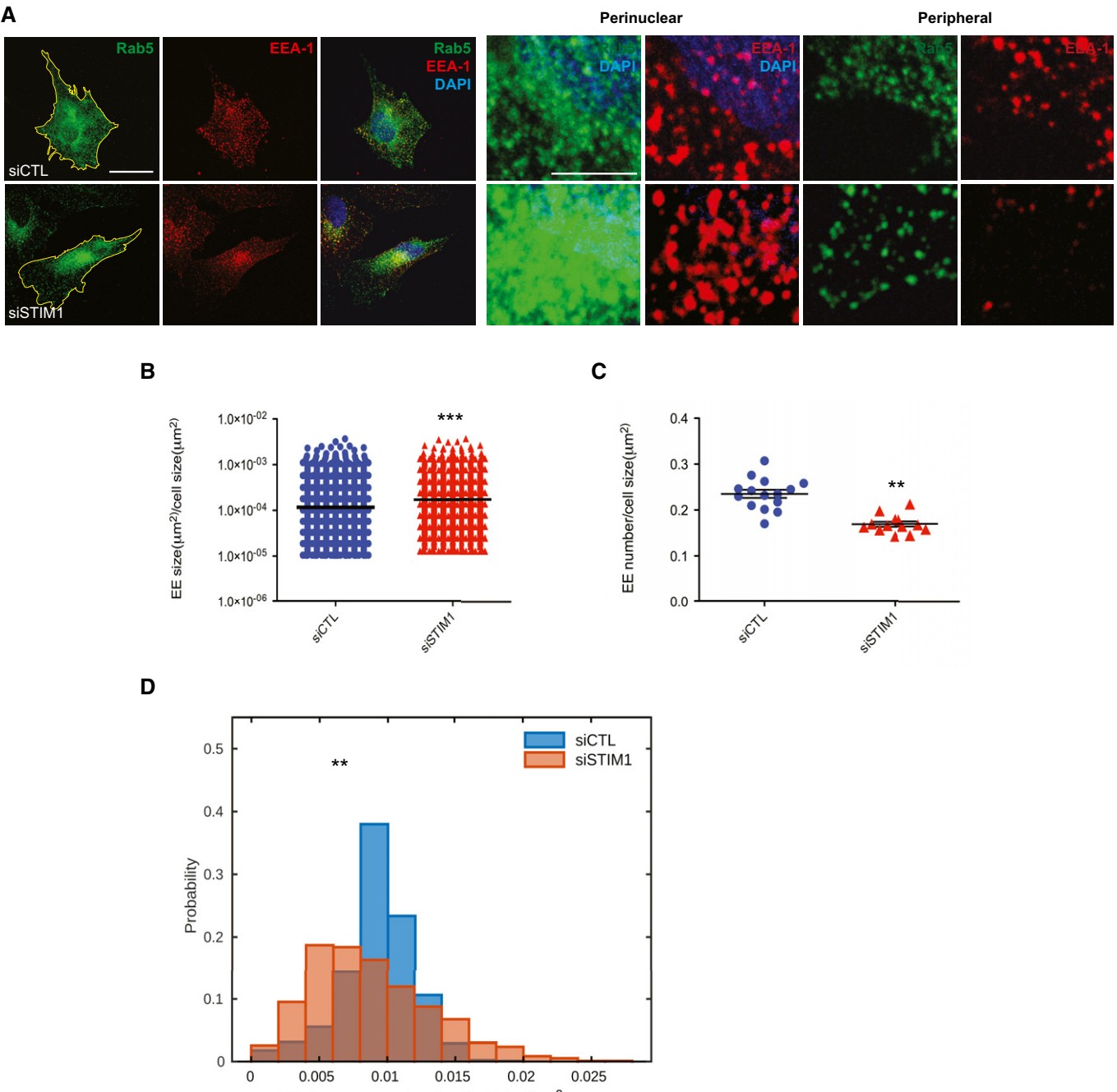

**Figure 3.**

**Figure 3.  STIM1 regulates early endosome retrograde transport.**

A  Confocal microscopy images of ECs silenced with a control siRNA (siCTL) or one targeting STIM1 (siSTIM1) and stained for endogenous Rab5 (in green) and EEA-1 (in red) to visualize EEs and DAPI (in blue) to highlight the nucleus. The yellow line is drawn to define cell periphery. Scale bar = 20 μm. On the right, inset panels to highlight respective perinuclear and peripheral area of the cell. Scale bar = 5 μm.

B  Average size, normalized on cell size, of EEA-1[+] endosomes as in A. Results are the average ± SEM of three independent experiments for a total 4,483 early endosomes in 18 siCTL cells (299 ± 33 endosomes per cell) and 2,454 early endosomes in 13 siSTIM1 cells (189 ± 17 endosomes per cell) and analyzed by a two-tailed heteroscedastic Student's *t*-test, $P \leq 0.001$***.

C  Average number, normalized on cell size, of EEA-1[+] endosomes as in A. Results are the average ± SEM of three independent experiments for a total 4,483 early endosomes in 18 siCTL cells (299 ± 33 endosomes per cell) and 2,454 early endosomes in 13 siSTIM1 cells (189 ± 17 endosomes per cell) and analyzed by a two-tailed heteroscedastic Student's *t*-test, $P \leq 0.01$**.

D  Distribution of distance to nucleus at the first frame, normalized on cell size, quantified by image (data correspond to Movies EV1 and EV2) segmentation (see Materials and Methods, Live imaging experiments) of Rab5[+] endosomes, appearing for 60 s. Results are from three independent experiments for a total of 903 early endosomes in 18 siCTL cells (50 ± 5 endosomes per cell) and 935 early endosomes in 20 siSTIM1 cells (47 ± 9 endosomes per cell). Data are analyzed by a two-tailed heteroscedastic Student's *t*-test, $P \leq 0.01$**.

Source data are available online for this figure.

other cell types, such as Hs746T carcinoma cells (Fig EV2B, first two rows) and MRC5 fibroblasts (Fig EV2C, first two rows). These data suggest a cooperation between STIM1 and the dynein/dynactin complex in regulating the retrograde transport of both LEs and EEs.

## Concerted antagonistic action of STIM1/dynein and KIFC1 controls early endosome retrograde transport

Automated quantitative analyses of EE positioning in ECs revealed that CilioD-mediated inhibition of dynein decreased EEA-1[+] EE distance to nucleus (Fig 4F). These data imply that a further retrograde motor may antagonize STIM1/dynein, conceivably to finely tune the transport rate of these specific endosomal cargos toward the nucleus (Hu *et al*, 2017). In this regard, two different reciprocally antagonizing kinesin (KIF) 2 anterograde motors were reported to cooperate to move the same intraflagellar transport particles along MTs in the direction of the tip of cilia of *Caenorhabditis elegans* neurons (Pan *et al*, 2006). KIFs are major motor proteins that, tugging war with dynein (Ayloo *et al*, 2014; Reck-Peterson *et al*, 2018), primarily drive the anterograde transport of endosomes to MT plus ends (Hirokawa *et al*, 2009). Interestingly, KIF-14 family members are C-terminal motor proteins, which contain an EB1 binding SxIP motif (Braun *et al*, 2013) and move toward the minus end of MTs, thereby regulating retrograde cargo motion (She & Yang, 2017). In humans, this family comprises only three known members: KIFC1, KIFC2, and KIFC3. In particular, KIFC1, whose role in mitosis has been widely analyzed (Cross & McAinsh, 2014; Hepperla *et al*, 2014), has also been recently shown to be required for MT minus end-directed cargo traffic and cell behavior in either normal (Nath *et al*, 2007; Braun *et al*, 2016) or cancer cells (De *et al*, 2009; Grinberg-Rashi *et al*, 2009; Patel *et al*, 2018). Moreover, KIFC1 was previously found to be recruited on EEs (Mukhopadhyay *et al*, 2011). Additionally, land plants, which lack of dynein, evolved a large number of KIF-14s, likely to fill the functional roles left by dynein (Gicking *et al*, 2018). Therefore, we explored the possibility that KIFC motors may be involved in the EE retrograde transport along MT in ECs. We first measured the mRNA expression levels of the three KIFC family genes, found that the only one expressed is KIFC1 (Fig 5A), and silenced it, alone or in combination with STIM1 silencing (Fig 5B). Confocal and automatic quantitative analysis showed that, similarly to what observed upon STIM1 knock-down (Fig 3A and D) or dynein inhibition (Fig 4E and F), KIFC1 silencing

causes a dramatic concentration of enlarged EEA1[+] EEs in the perinuclear area of ECs (Fig 5C and D). However, differently from siSTIM1 (Fig 2A and D) or CilioD-treated cells (Fig 4E and G), the positioning of LEs was left completely unchanged in siKIFC1 ECs (Figs 5C and EV2D). Of note, the simultaneous silencing of KIFC1 and STIM1 resulted in more peripherally distributed EEA1[+] EEs, as seen for siCTL ECs (Fig 5C and E). Exploiting the KIFC1 ATPase-specific inhibitor AZ82 (Wu *et al*, 2013; Park *et al*, 2017), we similarly observed that the abolishment of the kinesin retrograde transport also diminished the distance of EE to the nucleus (Fig 6A and B), as seen for dynein activity diminishment (Fig 4F). Moreover, using the two inhibitors, alone or in combination, we further substantiated the antagonistic synergism between the two motors in driving EE, but not LE, retrograde transport in human ECs. Indeed, automated analysis of time-lapse confocal microscopy (Movies EV3–EV5) revealed a comparable augmented velocity in the retrograde motion of Rab5[+] EEs in ECs treated with either drugs (Fig 6C). Additionally, we detected equivalent changes in the distance to the nucleus of EE, but not LE, positioning in both Hs746T carcinoma cells (Fig EV2B, third row) and MRC-5 fibroblasts (Fig EV2C, third row), thus confirming a specific involvement of distinct motor sets for different endocytic vesicle motion in different cell types. Interestingly, as observed in double siKIFC1/siSTIM1 cells (Fig 5C and E), the increased retrograde transport of EEs in ECs treated with one drug only was abolished when CilioD-treated cells were simultaneously incubated with AZ82 (Movie EV6; Fig 6A, right panels, and Fig 6D). Hence, KIFC1 acts as a retrograde motor that antagonistically cooperates with STIM1/dynein to specifically and finely control the centripetal motion of EEs along MTs.

Considering the known role of KIF5B as anterograde kinesin modulating EE motility and traffic toward the MT plus ends (Nath *et al*, 2007; Loubéry *et al*, 2008; Braun *et al*, 2016), we evaluated whether this motor may regulate the centrifugal motion of EEA-1[+] EEs observed in our double retrograde motor silenced (Fig 5E) or inhibited (Fig 6D) ECs. Indeed, we simultaneously treated KIF5B silenced ECs with CilioD and AZ82 and analyzed EE positioning (Fig EV2E). Confocal fluorescence microscopy analysis unveiled how the silencing of KIF5B completely abolishes the peripheral localization of EEA-1[+] EEs observed upon concurrent stimulation of ECs with CilioD and AZ82 (Fig EV2E). Those data identify KIF5B as a key anterograde motor that counterbalances the activity of the

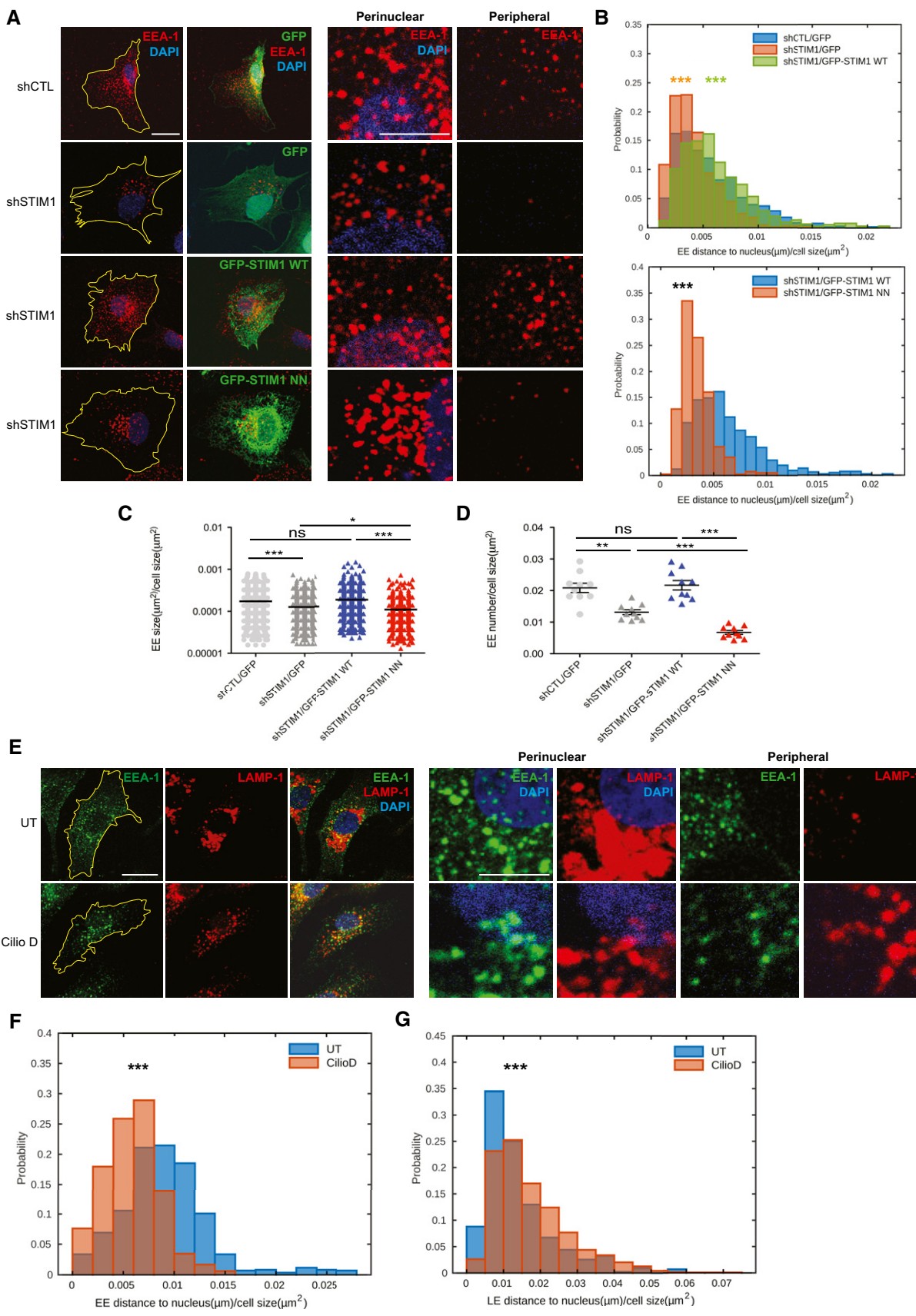

Figure 4.

Figure 4. STIM1 and dynein oppositely regulate early and late endosome retrograde transport.

A   Confocal microscopy images of ECs silenced with a control shRNA (shCTL) or one targeting STIM1 (shSTIM1), rescued for GFP expression alone (shCTL/GFP and shSTIM1/GFP) or for GFP-STIM1 WT (shSTIM1/GFP-STIM1 WT) or GFP-STIM1 NN (shSTIM1/GFP-STIM1 NN) and stained for endogenous EEA-1 (in red) to visualize EEs and DAPI (in blue) to highlight the nucleus. The yellow line is drawn to define cell periphery. Scale bar = 20 μm. On the right, inset panels to highlight respective perinuclear and peripheral area of the cell. Scale bar = 5 μm.

B   Upper panel, Distribution of distance to nucleus, normalized on cell size, quantified by image (as in A) segmentation (see Materials and Methods, Confocal microscopy and early/late quantification) of EEA-1$^+$ endosomes. Results are from two independent experiments for a total of 830 early endosomes in 13 shCTL/GFP cells (64 ± 6 endosomes per cell), 719 early endosomes in 10 shSTIM1/GFP cells (60 ± 6 endosomes per cell), and 820 early endosomes in 12 shSTIM1/GFP-STIM1 WT cells (68 ± 6 endosomes per cell) and analyzed by a parametric two-tailed analysis of variance (ANOVA) with Bonferroni post hoc analysis. ANOVA $P \leq 0.001$; Bonferroni for shCTL/GFP and shSTIM1/GFP $P \leq 0.001***$ (orange) and for shSTIM1/GFP and shSTIM1/GFP-STIM1 WT $P \leq 0.001***$ (green). Lower panel, Distribution of distance to nucleus, normalized on cell size, quantified by image (as in A) segmentation (see Materials and Methods, Confocal microscopy and early/late quantification) of EEA-1$^+$ endosomes. Results are from two independent experiments for a total of 820 early endosomes in 12 shSTIM1/GFP-STIM1 WT cells (68 ± 6 endosomes per cell) and 400 early endosomes in 12 shSTIM1/GFP-STIM1 NN cells (33 ± 7 endosomes per cell) and analyzed a two-tailed heteroscedastic Student's t-test, $P \leq 0.001***$.

C   Average size, normalized on cell size, quantified by image (as in A) segmentation (see Materials and Methods, Confocal microscopy and early/late quantification) of EEA-1$^+$ endosomes. Results are the average ± SEM of two independent experiments for a total 830 early endosomes in 13 shCTL/GFP cells (64 ± 6 endosomes per cell), 719 early endosomes in 10 shSTIM1/GFP cells (60 ± 6 endosomes per cell), 820 early endosomes in 12 shSTIM1/GFP-STIM1 WT cells (68 ± 6 endosomes per cell), and 400 early endosomes in 12 shSTIM1/GFP-STIM1 NN cells (33 ± 7 endosomes per cell) and analyzed by a parametric two-tailed analysis of variance (ANOVA) with Bonferroni post hoc analysis. ANOVA $P \leq 0.001$; Bonferroni for shCTL/GFP and shSTIM1/GFP $P \leq 0.001***$, for shCTL/GFP and shSTIM1/GFP-STIM1 WT $P > 0.05$ not significant (ns), for shSTIM1/GFP-STIM WT and shSTIM1/GFP-STIM1 NN $P \leq 0.001***$ and for shSTIM1/GFP and shSTIM1/GFP-STIM1 NN $P \leq 0.05*$.

D   Average number, normalized on cell size, quantified by image (as in A) segmentation (see Materials and Methods, Confocal microscopy and early/late quantification) of EEA-1$^+$ endosomes. Results are the average ± SEM of two independent experiments for a total 830 early endosomes in 13 shCTL/GFP cells (64 ± 6 endosomes per cell), 719 early endosomes in 10 shSTIM1/GFP cells (60 ± 6 endosomes per cell), 820 early endosomes in 12 shSTIM1/GFP-STIM1 WT cells (68 ± 6 endosomes per cell), and 400 early endosomes in 12 shSTIM1/GFP-STIM1 NN cells (33 ± 7 endosomes per cell) and analyzed by a parametric two-tailed analysis of variance (ANOVA) with Bonferroni post hoc analysis. ANOVA $P \leq 0.001$; Bonferroni for shCTL/GFP and shSTIM1/GFP $P \leq 0.01**$, for shCTL/GFP and shSTIM1/GFP-STIM1 WT $P > 0.05$ not significant (ns), for shSTIM1/GFP-STIM WT and shSTIM1/GFP-STIM1 NN $P \leq 0.001***$ and for shSTIM1/GFP and shSTIM1/GFP-STIM1 NN $P \leq 0.001***$.

E   Confocal microscopy images of untreated ECs (UT) or treated with Ciliobrevin D (CilioD) and stained for endogenous EEA-1 (in green) and LAMP-1 (in red) to visualize EEs and LEs, respectively, and DAPI (in blue) to highlight the nucleus. The yellow line is drawn to define cell periphery. Scale bar = 20 μm. On the right, inset panels to highlight respective perinuclear and peripheral area of the cell. Scale bar = 5 μm.

F   Distribution of distance to nucleus at the first frame, normalized on cell size, quantified by image (data correspond to Movies EV3 and EV4) segmentation (see Materials and Methods, Live imaging experiments) of Rab5$^+$ endosomes, appearing for 30 s, during Ciliobrevin D (CilioD) treatment. Results are from three independent experiments for a total of 853 early endosomes in six untreated (UT) cells (142 ± 22 endosomes per cell) and 657 early endosomes in nine CilioD cells (73 ± 23 endosomes per cell) and analyzed by a two-tailed heteroscedastic Student's t-test, $P \leq 0.001***$.

G   Distribution of distance to nucleus, normalized on cell size, quantified by image (as in E) segmentation (see Materials and Methods, Confocal microscopy and early/late quantification) of LAMP-1$^+$ endosomes. Results are from three independent experiments for a total of 432 late endosomes in 12 untreated (UT) cells (36 ± 9 endosomes per cell) and 1,184 late endosomes in 18 CilioD cells (66 ± 9 endosomes per cell) and analyzed by a two-tailed heteroscedastic Student's t-test, $P \leq 0.001***$.

Source data are available online for this figure.

dual dynein/KIFC1 retrograde motor system to control the positioning of EEs in living cells.

Next, we explored the molecular mechanisms by which dynein and KIFC1 may connect to EE cargos in ECs. First, we checked whether STIM1 may interact with KIFC1, but we could not immunoprecipitate the two proteins together (Fig EV3A), excluding possible physical interactions between the two retrograde motor systems. Among potential adaptors connecting dynein with EEs, the CC domain containing members of the HOOK adaptor family were found to play such a role in filamentous fungi (Bielska et al, 2014; Zhang et al, 2014). Although HOOK1 (Maldonado-Báez et al, 2013; Olenick et al, 2019) and HOOK3 (Kendrick et al, 2019; Siddiqui et al, 2019) have already been studied in both retrograde and anterograde motion of endosomes in mammalian cells, yet the specific molecular mechanisms by which those may differentially control the movement of vesicles along MTs are not entirely understood (Luiro et al, 2004; Xu et al, 2008; Maldonado-Báez et al, 2013; Olenick et al, 2019). Therefore, we investigated whether and how dynein and KIFC1 may associate with the HOOK proteins and also whether p150Glued, which is a known cargo anchor (Deacon et al, 2003) and processivity factor (Berezuk & Schroer, 2007) also for some KIFs, may regulate the motor switch in ECs. We observed that HOOK1 interacts with both dynein and p150Glued in ECs, but not

with KIFC1 (Fig 7A), which instead specifically associates with p150Glued, but not with other components of the dynactin complex (such as p50) and HOOK3 (Fig 7B). Hence, at least two distinct EE retrograde motor/adaptor complexes exist in ECs, namely dynein/dynactin/HOOK1 and KIFC1/dynactin/HOOK3.

To get further insights about the role of HOOK1 and HOOK3 in the dynactin-coordinated antagonism between dynein and KIFC1, we analyzed the localization of EEA1$^+$ EEs and LAMP-1$^+$ LEs in ECs silenced for HOOK1 (siHOOK1) or HOOK3 (siHOOK3) or both of them (Fig 7C). Confocal analysis revealed that, compared to siCTL cells, EEs were highly concentrated close to the nucleus in siHOOK1 (Fig 7D and E) or siHOOK3 ECs (Fig 7D and F), mimicking EE distribution observed in the single siSTIM1 or siKIFC1 cells (Figs 3D and 5D), while LE positioning (known to be RILP-dependent (Berezuk & Schroer, 2007)) was unaffected by HOOK silencing (Figs 7D and EV3B and C), as for KIFC1 silencing (Fig EV2D). Of note, simultaneous HOOK1 and HOOK3 silencing rescued the EE abnormalities present in single siHOOK1 (P-value EE distance siHOOK1 and siHOOK3 $P > 0.05$ not significant) ECs (Fig 7D and G), mimicking what observed with the double silencing or inhibition of the two-motor system (Figs 5C and E, and 6A and D). These data confirm a complementary role of the two, dynein/HOOK1 and KIFC1/HOOK3, motor/adaptor complexes in driving retrograde EE, and not LEs,

transport. To further validate the competition between those two-motor systems for the binding to p150Glued, we immunoprecipitated this protein together with the HOOK1 adaptor in untreated (UT) or AZ82-treated ECs (Fig 7H). Interestingly, we discovered that the AZ82-induced decrease in KIFC1 activity strengthens the binding

of HOOK1 to p150Glued (Fig 7H) that favors the dynein/HOOK1 motor/adaptor complex, in agreement with the increased EE retrograde motion observed in siKIFC1 ECs (Fig 5C and D). Moreover, concurring with our finding that STIM1 favors EB1-dynactin complex formation and dynein-driven retrograde transport (Figs 1

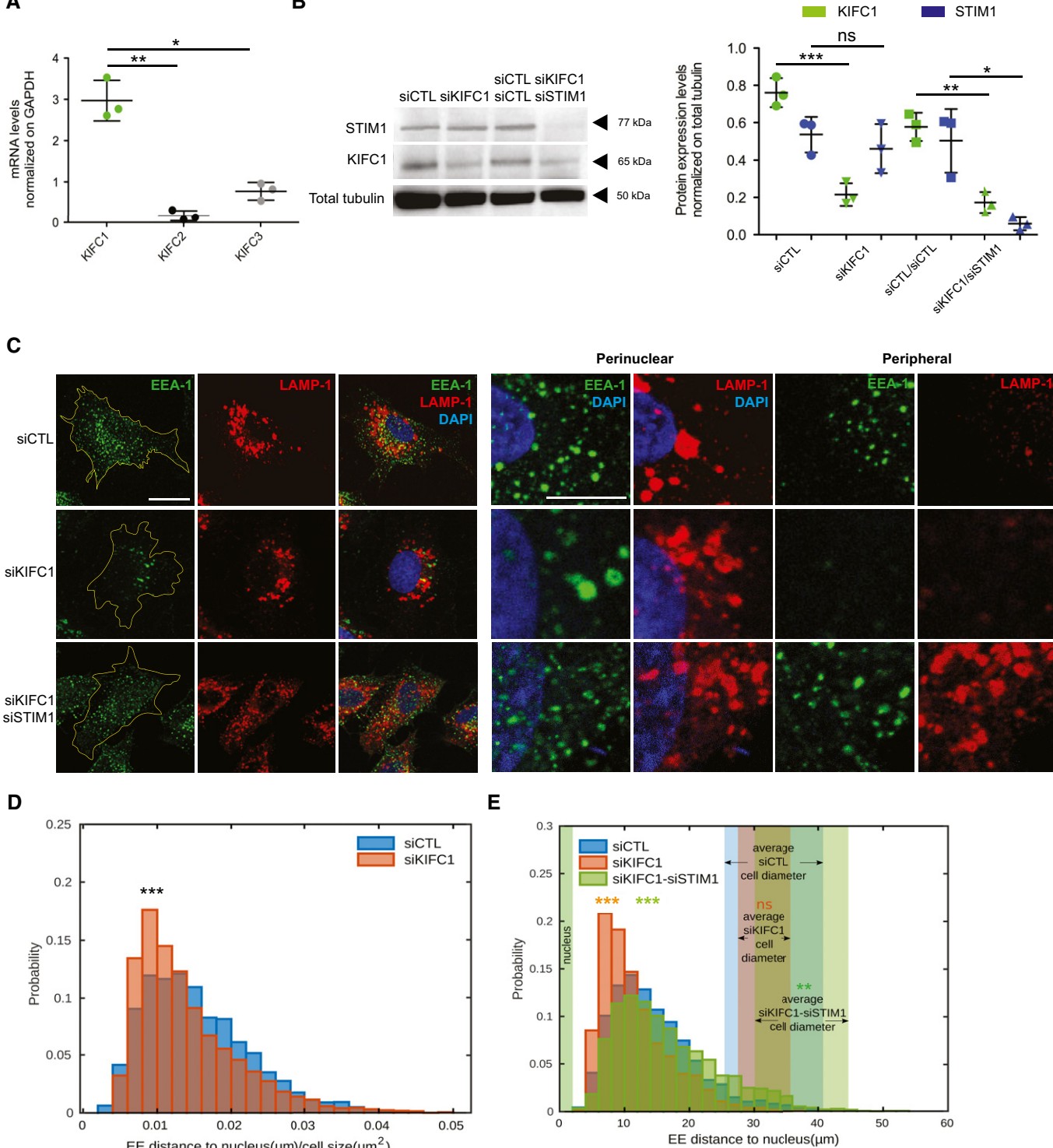

**Figure 5.**

**Figure 5. KIFC1 acts as an additional early endosome-dedicated MT retrograde motor.**

A  mRNA levels of KIFC1, KIFC2, and KIFC3 in ECs. Values are normalized on GAPDH mRNA levels. Results are the average $\pm$ SD of three independent experiments and analyzed by a parametric two-tailed analysis of variance (ANOVA) with Bonferroni *post hoc* analysis. ANOVA $P \leq 0.01$; Bonferroni for KIFC1 and KIFC2 $P \leq 0.01**$ and for KIFC1 and KIFC3 $P \leq 0.05*$.

B  Representative Western blot analysis of endogenous KIFC1 and STIM1 proteins (both normalized on total tubulin levels) in ECs silenced with a control siRNA (siCTL) or one targeting KIFC1 (siKIFC1) or two control siRNAs (siCTLsiCTL) or two targeting KIFC1 and STIM1 (siKIFC1siSTIM1). Results are the average $\pm$ SD of three independent experiments and analyzed by a parametric two-tailed analysis of variance (ANOVA) with Bonferroni *post hoc* analysis. ANOVA $P \leq 0.01$; Bonferroni for siCTL and siKIFC1, KIFC1 expression $P \leq 0.001***$, STIM1 expression $P > 0.05$ not significant (ns) and for siCTLsiCTL and siKIFC1siSTIM1, KIFC1 expression $P \leq 0.01**$, STIM1 expression $P \leq 0.05*$.

C  Confocal microscopy images of ECs silenced with a control siRNA (siCTL) or one targeting KIFC1 (siKIFC1) or one targeting STIM1 (siSTIM1) or two targeting both (siKIFC1siSTIM1) and stained for endogenous EEA-1 (in green) and LAMP-1 (in red) to visualize EEs and LEs, respectively, and DAPI (in blue) to highlight the nucleus. The yellow line is drawn to define cell periphery. Scale bar = 20 μm. On the right, inset panels to highlight respective perinuclear and peripheral area of the cell. Scale bar = 5 μm.

D  Distribution of distance to nucleus, normalized on cell size, quantified by image (as in C) segmentation (see Materials and Methods, Confocal microscopy and early/late quantification) of EEA-1$^+$ endosomes in siCTL and siKIFC1 ECs. Results are from three independent experiments for a total of 4,091 early endosomes in 19 siCTL cells (215 $\pm$ 20 endosomes per cell) and 2,275 early endosomes in 20 siKIFC1 cells (114 $\pm$ 10 endosomes per cell) and analyzed by a two-tailed heteroscedastic Student's *t*-test, $P \leq 0.001***$.

E  Distribution of distance to nucleus quantified by image (as in C) segmentation (see Materials and Methods, Confocal microscopy and early/late quantification) of EEA-1$^+$ endosomes. Results are from three independent experiments for a total of 4,091 early endosomes in 19 siCTL cells (215 $\pm$ 20 endosomes per cell), 2,275 early endosomes in 20 siKIFC1 cells (114 $\pm$ 10 endosomes per cell), and 3,257 early endosomes in 18 siKIFC1/siSTIM1 cells (181 $\pm$ 14 endosomes per cell). Data are analyzed by a parametric two-tailed analysis of variance (ANOVA) with Bonferroni *post hoc* analysis. ANOVA $P \leq 0.001$; Bonferroni for siCTL and siKIFC1 ECs $P \leq 0.001***$ (orange) and for siKIFC1 and siSTIM1/siKIFC1 ECs $P \leq 0.001***$ (green). On the right hand side of the plot, colored bands represent the average cell diameter, significantly changing throughout the conditions. Data are analyzed by a parametric two-tailed analysis of variance (ANOVA) with Bonferroni *post hoc* analysis. ANOVA $P \leq 0.01$; Bonferroni for siCTL and siKIFC1 ECs $P > 0.05$ not significant (orange ns) and for siKIFC1 and siSTIM1/siKIFC1 ECs $P \leq 0.01**$ (green).

Source data are available online for this figure.

and 2), we also found that the lack of STIM1 increased the binding of KIFC1 to EB1 in ECs (Fig EV3D). Altogether, these data sustain a two-layered competition model between STIM1/dynein and KIFC1 to interact with the p150Glued subunit of dynactin and EB-1 to cooperatively regulate EE retrograde transport.

## STIM1 controls late endosome pH, mTORC1 signaling, and autophagy

Mounting evidences show how perinuclear rather than peripheral subcellular localization crucially influences the functioning of LEs (Ballabio & Bonifacino, 2020). For instance, the low luminal pH of perinuclear, but not peripheral LEs, favors the enzymatic activity of lysosomal hydrolases and the secondary active transport of different substrates by lysosomal transporters (Gowrishankar & Ferguson, 2016; Johnson *et al*, 2016). As another example, under nutrient-rich conditions, nutrient-regulated mechanistic target of rapamycin complex 1 (mTORC1), which signals to promote cell growth and to inhibit autophagy (Liu & Sabatini, 2020), is activated on the cytosolic surface of peripheral, but not perinuclear LEs (Korolchuk *et al*, 2011). Therefore, since LE positioning has been reported to be critical in the regulation of key cellular functions, we sought to investigate whether STIM1 may also impact on some of those, such as luminal pH, mTORC1 signaling, and autophagy. To measure the luminal pH of LEs in both control and ECs silenced for STIM1, we exploited the acetoxymethyl (AM) ester-modified fluorogenic intracellular pH probe pHrodo Green AM (Kulkarni *et al*, 2019), weakly fluorescent at neutral pH and whose fluorescence proportionally increases as pH lowers (Fig 8A). The relative amount of pHrodo Green AM-labeled acidic LAMP1$^+$ LEs in shSTIM1 cells decreased significantly compared to controls (Fig 8B), showing the peripheral LEs, observed only in ECs lacking for STIM1, to be much less marked by the fluorescent dye and therefore much less acidic. Considering the reported greater activation of mTORC1 signaling on

the cytosolic surface of peripheral endosomes (Korolchuk *et al*, 2011), next we measured the expression of mTORC1 scaffold protein regulatory-associated protein Raptor and the phosphorylation of the anabolic downstream mTOR effector p70 S6 kinase 1 (S6K1) (Liu & Sabatini, 2020), in control or STIM1 silenced ECs, enriched with those peripheral LEs. Although Raptor expression was not affect, we observed that STIM1 silencing significantly increased S6K1 phosphorylation (Fig 8C). Since mTORC1 also signals to inhibit catabolic processes such as autophagy, we evaluated whether STIM1 may also regulate this multistep process (Hansen *et al*, 2018). Phagophore formation and expansion is a key step in autophagy in which MT-associated protein light chain 3 (LC3) is first cleaved by the protease ATG4 to generate LC3-I (Hansen *et al*, 2018). Next, LC3-I is conjugated by ATG3 with phosphatidylethanolamine to form LC3-II that is then incorporated into autophagosomal membranes to interact with LC3-interacting motif bearing cargo receptors (Hansen *et al*, 2018). When we analyzed by Western blot LC3-I and -II protein levels, we noted that, compared to control, STIM1 silencing decreased the most relevant LC3-II product, indicating a defected autophagic flux as evaluated by the LC3-II/LC-3I ratio (Fig 8D). Altogether, those data confirm that STIM1 significantly impacts on different critical aspects of cellular function that are known to rely on LE subcellular localization (Korolchuk *et al*, 2011; Johnson *et al*, 2016; Ballabio & Bonifacino, 2020) that we discovered to be controlled by STIM1 itself.

## Discussion

In mammalian cells, the cytosolic positioning of endosomes crucially influences their functions and the fate of trafficked proteins (Bonifacino & Neefjes, 2017; Neefjes *et al*, 2017). The distance of endosomes from the PM correlates with different acidity (Mellman *et al*, 1986; Huotari & Helenius, 2011), proteolytic activity

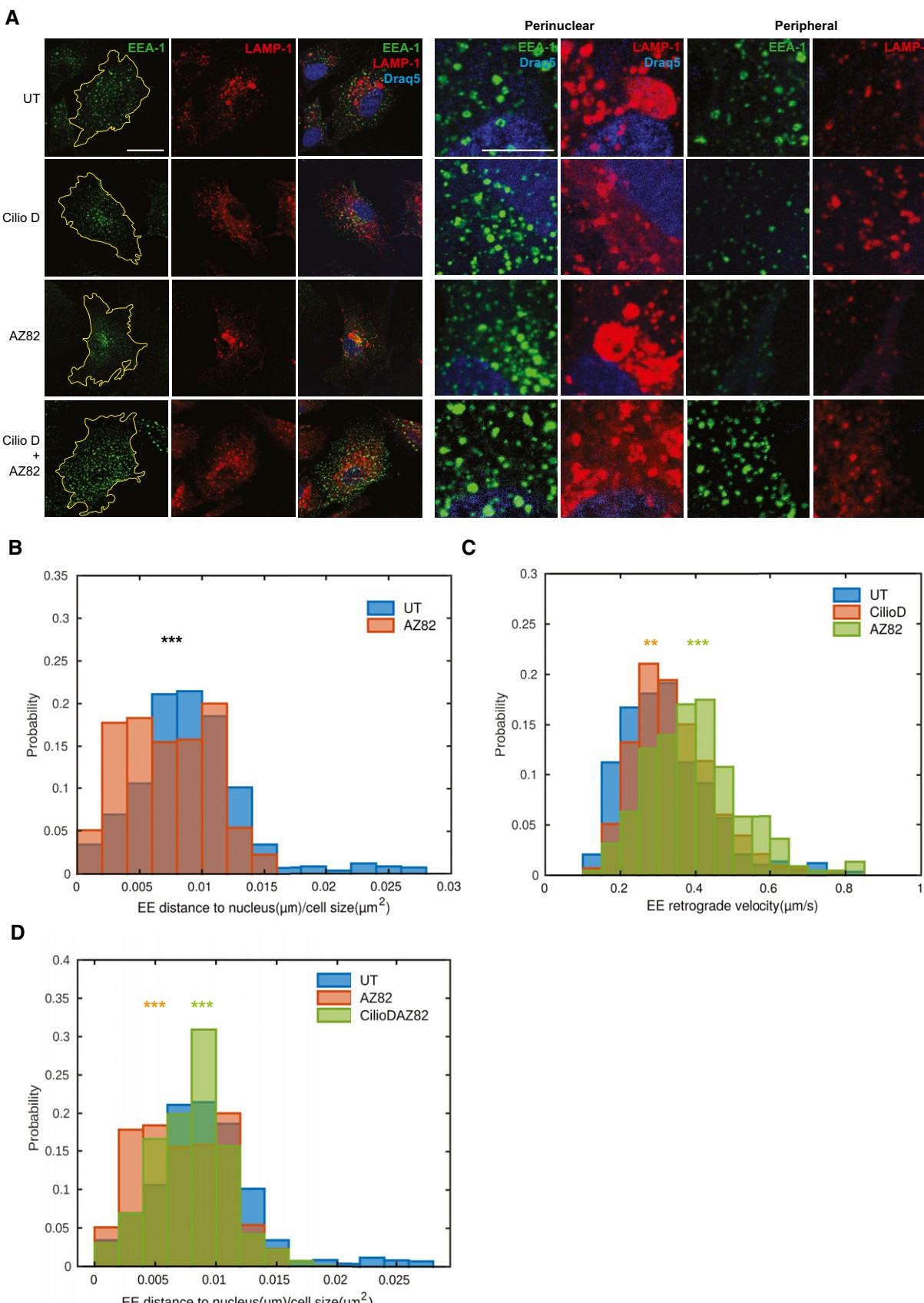

Figure 6.

◄

**Figure 6. Concerted antagonistic action of dynein and KIFC1 controls early endosome retrograde transport.**

A   Confocal microscopy images of untreated (UT) or treated with Ciliobrevin D (CilioD) or AZ82 or both treatments simultaneously (CilioD + AZ82) ECs and stained for endogenous EEA-1 (in green) and LAMP-1 (in red) to visualize EEs and LEs, respectively, and Draq5 (in blue) to highlight the nucleus. The yellow line is drawn to define cell periphery. Scale bar = 20 μm. On the right, inset panels to highlight respective perinuclear and peripheral area of the cell. Scale bar = 5 μm.

B   Distribution of distance to nucleus at the first frame, normalized on cell size, quantified by image (data correspond to Movies EV3 and EV5) segmentation (see Materials and Methods, Live imaging experiments) of Rab5$^+$ endosomes, appearing for 30 s, during AZ82 treatment in ECs, compared to untreated cells (UT). Results are from two independent experiments for a total of 853 early endosomes in six untreated cells (142 ± 22 endosomes per cell) and 355 early endosomes in 7 AZ82 cells (51 ± 12 endosomes per cell). Data are analyzed by a two-tailed heteroscedastic Student's *t*-test, $P \leq 0.001^{***}$.

C   Distribution of retrograde velocity, quantified by image (data correspond to Movies EV3–EV5) segmentation (see Materials and Methods, Live imaging experiments) of Rab5$^+$ endosomes, appearing for 30 s, during Ciliobrevin D (CilioD D) or AZ82 treatment in ECs, compared to untreated cells (UT). Results are from three independent experiments for a total of 580 EEs in six UT cells (97 ± 18 endosomes per cell), 432 EEs in nine CilioD cells (48 ± 15 endosomes per cell) and from two independent experiments for a total of 223 EEs in seven AZ82 cells (32 ± 6 endosomes per cell). Data are analyzed by a parametric two-tailed analysis of variance (ANOVA) with Bonferroni *post hoc* analysis. ANOVA $P \leq 0.001$; Bonferroni for UT and CilioD-treated ECs $P \leq 0.01^{**}$ and for UT and AZ82 treated ECs $P \leq 0.001^{***}$.

D   Distribution of distance to nucleus at the first frame, normalized on cell size, quantified by image (data correspond to Movies EV3, EV5 and EV6) segmentation (see Materials and Methods, Live imaging experiments) of Rab5$^+$ endosomes, appearing for 30 s, during AZ82 treatment (as in B), alone or in combination with CilioD treatment (CilioD + AZ82), compared to untreated EC cells (UT). Results are from two independent experiments for a total of 853 early endosomes in six UT cells (142 ± 22 endosomes per cell), 355 EEs in seven AZ82 cells (51 ± 12 endosomes per cell), and 1,106 EEs in eight CilioD + AZ82 cells (136 ± 42 endosomes per cell) and analyzed by a parametric two-tailed analysis of variance (ANOVA) with Bonferroni *post hoc* analysis. ANOVA $P \leq 0.001$; Bonferroni for UT and AZ82 treated ECs $P \leq 0.001^{***}$ (orange) and for CilioD and CilioD + AZ82 treated ECs $P \leq 0.001^{***}$ (green).

Source data are available online for this figure.

(Gowrishankar *et al*, 2015; Johnson *et al*, 2016), recycling kinetics of internalized receptors (Wilson *et al*, 2018), mTORC1 activation, and autophagic flux (Korolchuk *et al*, 2011; Ballabio & Bonifacino, 2020). Subcellular distribution also dictates the identity of lipids and proteins bidirectionally trafficked between endosomes and perinuclear or peripheral organelles, such as the Golgi apparatus (De Matteis & Luini, 2008; Progida & Bakke, 2016) or the tubular ER network (Wu *et al*, 2018), respectively. Yet, the molecular mechanisms that in mammalian cells control the different subcellular positioning of prototypic functionally distinct classes of endosomes, such EEs and LEs, are unknown. Here, we unveil that the starkly

diverse steady-state distribution of EEs and LEs/LYs relies on the employment of two distinct retrograde motor sets.

We discovered that while, as well-known (Bonifacino & Neefjes, 2017; Neefjes *et al*, 2017), dynein is the only MT minus end-directed motor responsible for LE/LY retrograde traffic, the transport of EEs toward the nucleus requires instead a competitive synergism of dynein and KIFC1 MT retrograde motors. Indeed, we found that CilioD-mediated inhibition of dynein, while dispersing smaller LEs throughout the cytoplasm, surprisingly increases the centripetal displacement velocity of EEs that appeared instead enlarged and clustered in the perinuclear area.

◄

**Figure 7. Dynein and KIFC1 control early endosome retrograde transport via distinct HOOK adaptors and by competing for the binding to p150Glued.**

A   Representative of three Western blot analysis of endogenous light-intermediate chain (LIC) of cytoplasmic dynein 1, KIFC1, and p150Glued immunoprecipitated with HOOK1 in wild-type ECs. Negative control (CTL) was performed incubating cell lysate with protein A- or G-Sepharose and empty rabbit IgG.

B   Representative of three Western blot analysis of endogenous p150Glued, HOOK3, and p50 (or Dynamitin) immunoprecipitated with KIFC1 in wild-type ECs. Negative control (CTL) was performed incubating cell lysate with protein A- or G-Sepharose and empty mouse IgG.

C   Representative Western blot analysis of endogenous HOOK1 or HOOK3 and total tubulin proteins in ECs, silenced with a control siRNA (siCTL) or one targeting HOOK1 (siHOOK1; on the left) or one targeting HOOK3 (siHOOK3; on the right).

D   Confocal microscopy images of ECs silenced with a control siRNA (siCTL) or one targeting HOOK1 (siHOOK1) or one targeting HOOK3 (siHOOK3) or two targeting both (siHOOK1siHOOK3) and stained for endogenous EEA-1 (in green) and LAMP-1 (in red) to visualize EEs and LEs, respectively, and DAPI (in blue) to highlight the nucleus. The yellow line is drawn to define cell periphery. Scale bar = 20 μm. On the right, inset panels to highlight respective perinuclear and peripheral area of the cell. Scale bar = 5 μm.

E   Distribution of distance to nucleus, normalized on cell size, quantified by image (as in D) segmentation (see Materials and Methods, Confocal microscopy and early/late quantification) of EEA-1$^+$ endosomes. Results are from three independent experiments for a total of 1,193 early endosomes in 12 siCTL cells (99 ± 9 endosomes per cell) and 1,104 early endosomes in 21 siHOOK1 cells (53 ± 4 endosomes per cell) and analyzed by a two-tailed heteroscedastic Student's *t*-test, $P \leq 0.001^{***}$.

F   Distribution of distance to nucleus, normalized on cell size, quantified by image (as in D) segmentation (see Materials and Methods, Confocal microscopy and early/late quantification) of EEA-1$^+$ endosomes. Results are from three independent experiments for a total of 1,193 early endosomes in 12 siCTL cells (99 ± 7 endosomes per cell) and 1,282 early endosomes in 22 siHOOK3 cells (58 ± 5 endosomes per cell) and analyzed by a two-tailed heteroscedastic Student's *t*-test, $P \leq 0.001^{***}$.

G   Distribution of distance to nucleus quantified by image (as in D) segmentation (see Materials and Methods, Confocal microscopy and early/late quantification) of EEA-1$^+$ endosomes. Results are from three independent experiments for a total of 1,193 early endosomes in 12 siCTL cells (99 ± 7 endosomes per cell), 1,104 EEs in 21 siHOOK1 cells (53 ± 4 endosomes per cell), and 2,340 EEs in 19 siHOOK1/siHOOK3 cells (123 ± 11 endosomes per cell) and analyzed by a parametric two-tailed analysis of variance (ANOVA) with Bonferroni *post hoc* analysis. ANOVA $P \leq 0.001$; Bonferroni for siCTL and siHOOK1 ECs $P \leq 0.001^{***}$ (orange) and for siHOOK1 and siHOOK1/siHOOK3 ECs $P \leq 0.001^{***}$ (green). On the right hand side of the plot, colored bands represent the average cell diameter, significantly changing throughout the conditions. Data are analyzed by a parametric two-tailed analysis of variance (ANOVA) with Bonferroni *post hoc* analysis. ANOVA $P \leq 0.001$; Bonferroni for siCTL and siHOOK1 ECs $P > 0.05$ not significant (orange ns) and for siHOOK1 and siHOOK1/siHOOK3 ECs $P \leq 0.001^{***}$ (green).

H   Representative Western blot analysis of the endogenous p150Glued co-immunoprecipitated with HOOK1 in untreated (UT) and AZ82-treated ECs (left) and its quantification by normalized densitometry (right). Negative control (CTL) was performed incubating cell lysate with protein A- or G-Sepharose and empty rabbit IgG. Results are the average ± SD of three independent assays. UT value of each biological replicate was normalized on itself and so AZ82 experimental value. Results were analyzed by a parametric two-tailed analysis of variance (ANOVA) with Bonferroni *post hoc* analysis. ANOVA $P \leq 0.001$; Bonferroni for UT and AZ82 $P \leq 0.05^*$.

Source data are available online for this figure.

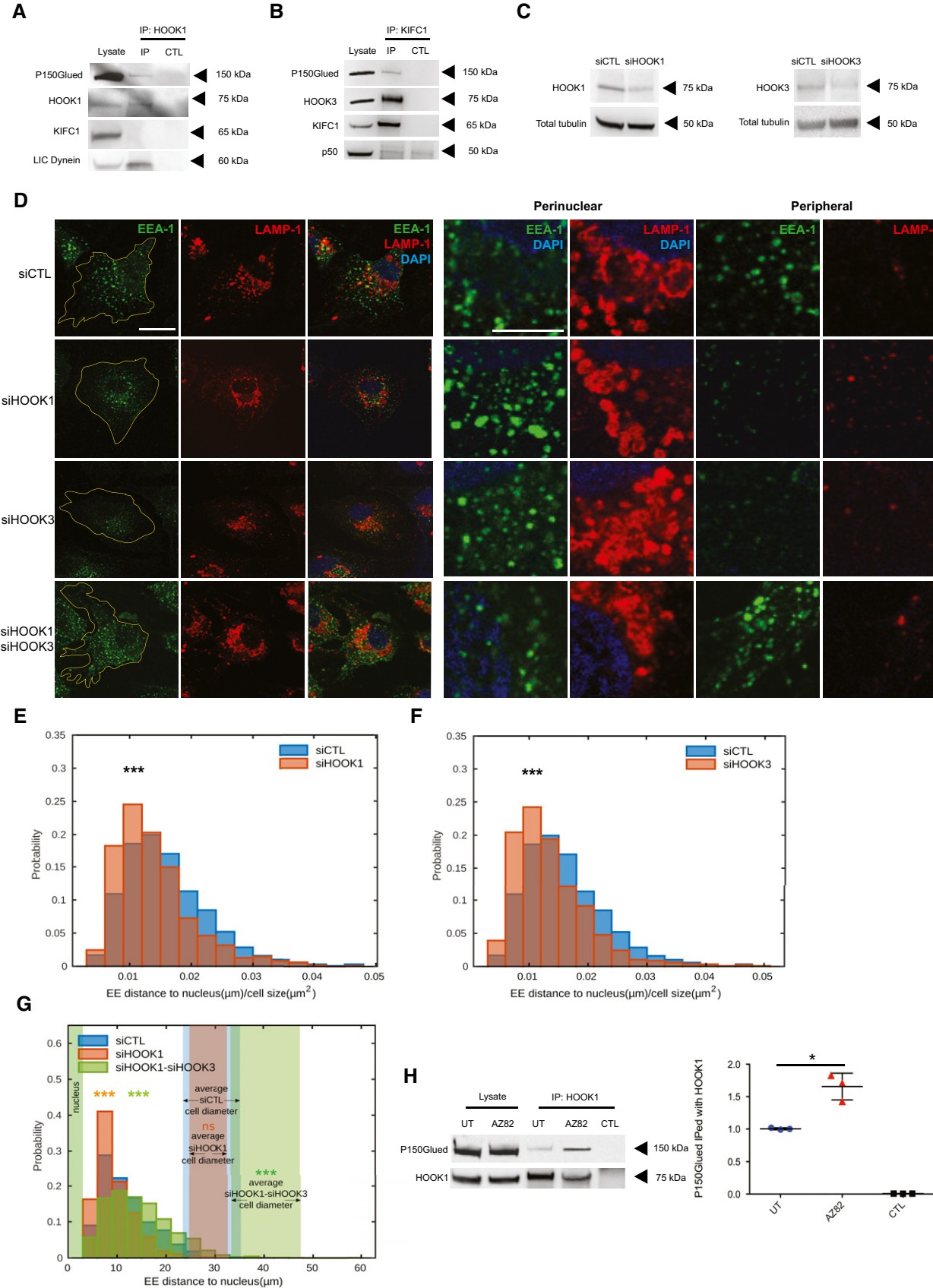

**Figure 7.**

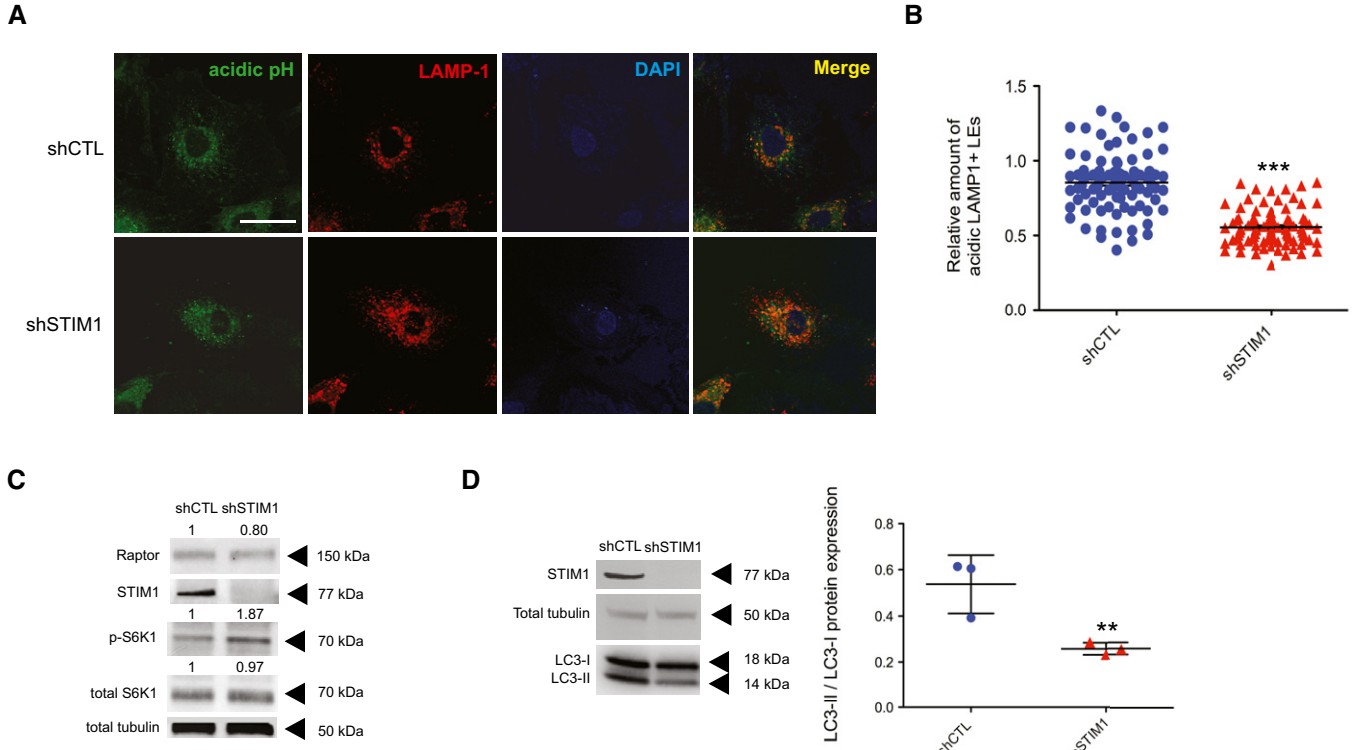

**Figure 8. STIM1 controls late endosome pH, mTORC1 signaling and autophagy.**

A  Confocal microscopy images of ECs silenced with a control shRNA (shCTL) or one targeting STIM1 (shSTIM1), to which the pH-sensitive dye pHrodo Green was given, and then, cells were stained for LAMP-1 (in red) to visualize LEs and DAPI (in blue) to highlight the nucleus. Scale bar = 20 μm.

B  Relative amount of LAMP-1$^+$ LEs positive to the pH-sensitive dye pHrodo (as in A) in shCTL and shSTIM1 ECs. Results are the average ± SEM of three independent experiments for a total of 90 cells (30 cell for experiment) and analyzed by a two-tailed heteroscedastic Student's t-test, $P \leq 0.001$ ***.

C  Representative Western blot analysis of endogenous Raptor, total and phospho-Thr389 S6K1 proteins (normalized on total tubulin levels) in ECs silenced with a control shRNA (shCTL) or one targeting STIM1 (siSTIM1). Numbers above each band represent respective N.O.D. average (two biological replicates) quantification.

D  Representative Western blot analysis of endogenous LC3-I and -II proteins (both normalized on total tubulin levels and represented as ratio LC3-II/LC3-I) in ECs silenced with a control shRNA (shCTL) or one targeting STIM1 (siSTIM1). Results are the average ± SD of three independent experiments and analyzed by a two-tailed heteroscedastic Student's t-test, $P \leq 0.01$**.

Source data are available online for this figure.

Our findings implied the existence of another retrograde motor synergistically antagonizing dynein that, by silencing and employing the specific inhibitor AZ82 (Wu *et al*, 2013; Park *et al*, 2017), we identified as the KIF-14 family member KIFC1 (She & Yang, 2017). Therefore, to retain their physiological small size and diffuse cytosolic distribution, EEs depend on dynein cooperative inhibition by KIFC1. Accordingly, LEs, which only rely on dynein, are larger and crowded around the nucleus; moreover, dynein inhibition by CilioD disperses LEs in the cytoplasm, substantially reducing their size. Many evidences on the role of motor-driven positioning of vesicles determining their size and number have indeed already being shown (Aoyama *et al*, 2017; Bonifacino & Neefjes, 2017; Hu *et al*, 2017). Our findings are in agreement with the funnel model (Collinet *et al*, 2010), according to which, as they move from the cell periphery to the center, endosomes progressively grow in size due to homotypic fusion. It appears that, in addition to determining their subcellular steady-state positioning, the velocity at which endosomes are transported along MTs toward the nucleus promotes their fusion,

likely due to the spatial convergence of MTs at MTOC. Of note, the observation that the retrograde transport of Rab5$^+$ endosomes along lengthy and slender cellular projections, such as neuronal axons, relies on dynein only (Guo *et al*, 2016) suggests that motor systems with different complexities are likely required to carry different vesicles in morphologically and functionally distinct cell types and structures.

Functional interactions such as tug-of-war and codependence between anterograde and retrograde MT motors (Fu & Holzbaur, 2014) or cooperative teamwork among same direction MT motors (Mallik *et al*, 2013) are well established. Mechanical competition between MT plus end-directed KIFs was also reported in *C. elegans* neuronal cilia (Pan *et al*, 2006). Here, we show that a mutual inhibition between dynein and KIFC1 drives EE retrograde transport. This result indicates that cooperative antagonism among MT motors oriented in the same direction may represent a further molecular strategy to finely regulate directional cargo transport in cells. The organization of endosomes in plants displays unique features distinct from those of animal cells, the trans-Golgi network

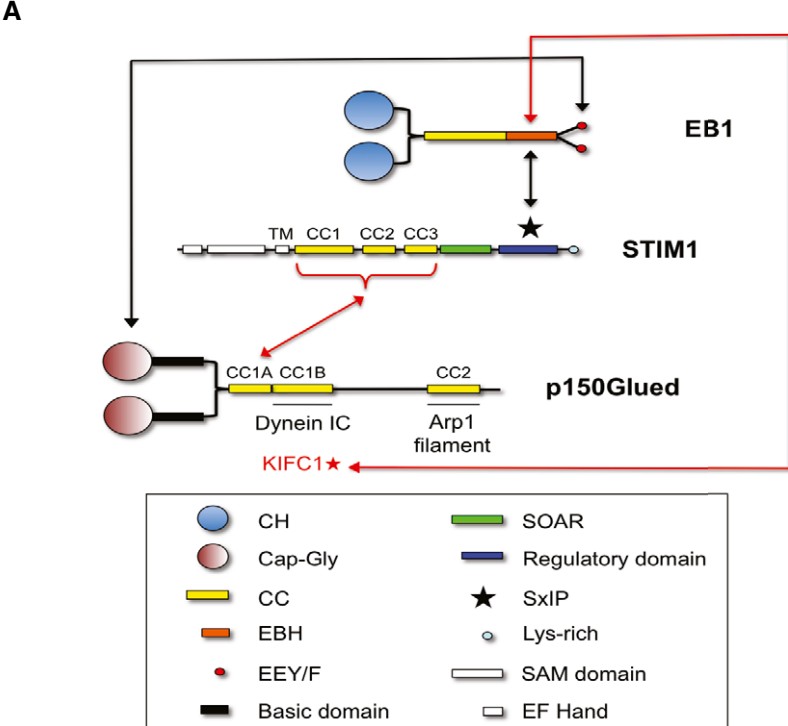

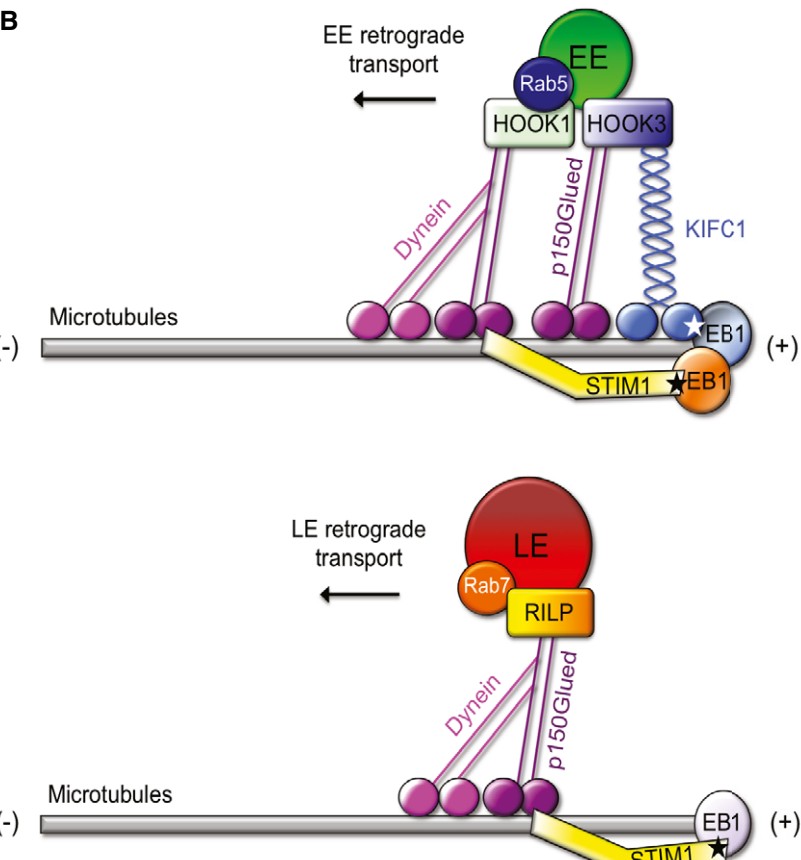

**Figure 9.**

**Figure 9.  Dynein and KIFC1 motor sets differentially drive p150Glued/STIM1-dependent early and late endosome retrograde transport along MTs.**

A   Schematic model of the molecular interactions between dynein/p150Glued/STIM1 and KIFC1/p150Glued complexes at the MT plus ends. Black arrows point at already known binding motifs, whereas red parts show interaction domains found in this manuscript. TM, transmembrane domain.

B   Schematic model of the distinct motor sets regulating EE (above) and LE (below) retrograde transport, along MTs. Stars indicate EB1-binding SxIP motifs.

Source data are available online for this figure.

functioning for example as an equivalent of EEs (Contento & Bassham, 2012). Our data suggest that, diverging from plants that lacked dynein and expanded the KIF-14 family (Gicking *et al*, 2018), the maintenance of both MT minus end-directed motor families may have allowed animal cells to evolve a differently organized and more sophisticated endolysosomal system. Furthermore, our findings suggest that the EE transport toward the MT plus ends, observed here in several conditions whereby the identified dual-motor retrograde movement system is inhibited, relies on the anterograde kinesin KIF5B (Nath *et al*, 2007; Loubéry *et al*, 2008; Braun *et al*, 2016).

We found that the interaction of the SxIP motif of the ER transmembrane protein STIM1 with the EBH domain of the MT plus end interacting protein EB1 (Grigoriev *et al*, 2008) is crucial for the dynein-dependent retrograde transport of both LEs and EEs in living cells, suggesting a dual role in the regulation of endosome positioning and $Ca^{2+}$ intra-organelle dynamics (Yuan *et al*, 2009; Vaca, 2010; Chang *et al*, 2018). Indeed, both indirectly, via the SxIP-enabled association to EB1, and directly, through its CC domains, STIM1 interacts with and clusters p150Glued via the CC1a motif, a crucial domain of the dynein activator complex dynactin (Zhang *et al*, 2017; McKenney, 2018), at the plus ends of MTs. The CC1a of p150Glued is known to exist in an inhibited state, folded with the second CC1b domain, masking the Cap-Gly motif and binding to MTs (Wang *et al*, 2014; Saito *et al*, 2020). In the resulting triple complex, a key function of STIM1 is that of stabilizing the contact between EB1 and the p150Glued subunit of dynactin, which the silencing of STIM1 severely hampers. Therefore, even if the CAP-Gly domain of p150Glued can bind the C-terminal EEY motif of EB1 (Akhmanova & Steinmetz, 2015; Rupam & Surrey, 2018), the ability of STIM1 to interact with both proteins significantly strengthens their association in living cells, perhaps by competing with the CC1b motif for the binding with the CC1a, supporting the model according to which functional networks, at MT plus ends, arise from combinations of several moderate/low-affinity protein–protein interactions (Akhmanova & Steinmetz, 2015). The finding that STIM1 silencing phenocopies the CilioD-mediated dynein inhibition effect on EE and LE steady-state distribution, without affecting their maturation, sustains the notion that STIM1 interacts with and positively regulates the dynein activating function of p150Glued (Zhang *et al*, 2017; McKenney, 2018). Analogously to the dynein/dynactin activator STIM1 (Fig 9A), KIFC1 also contains a SxIP motif, which allows its binding to EB1 and localization at the plus end of MTs (Braun *et al*, 2013), competing, as reported here, with STIM1. Furthermore, we found that KIFC1 interacts with p150Glued as well. It is hence conceivable that the appearance during evolution of an SxIP motif in KIFC1 enabled to coordinate the EB1-dependent recruitment of p150Glued/KIFC1 at MT plus ends with that of dynein/p150Glued/STIM1, allowing physiological MT minus end-directed transport of EEs in animal cells. The observation that HOOK1 and HOOK3 act as distinct and functionally counteracting adaptors for dynein-dependent and KIFC1-dependent, respectively,

MT minus end movement of EEs, further uphold a model of a two-layered (at MT ends and p150Glued/molecular adaptor levels) cooperative antagonism between same direction motors, involved in the retrograde transport of EEs along MTs (Fig 9B, above), but not LEs (Fig 9B, below).

High KIFC1 mRNA or protein levels have been detected in different human cancer types and found to correlate with poor prognosis (Grinberg-Rashi *et al*, 2009; Pannu *et al*, 2015; Ogden *et al*, 2017; Patel *et al*, 2018). In general, these data were interpreted in light of the KIFC1 ability to crosslink and slides MTs in mitotic spindles, enabling tight pole focusing (Cross & McAinsh, 2014; Hepperla *et al*, 2014). Indeed, the survival of cancer cells with supernumerary centrosomes was found to rely on KIFC1 ability to cluster them (Kwon *et al*, 2008; De *et al*, 2009; Kley-lein-Sohn *et al*, 2012). Here, we identified a crucial role of KIFC1 in controlling EE retrograde transport and dynamic distribution in both normal and cancer cells. Aberrant expression or function of proteins involved in the regulation of endosomal traffic has been involved in cancer development and progression (Goldenring, 2013; Lanzetti & Di Fiore, 2017). Thereby, the hypothesis by which KIFC1 overexpression may promote cancer growth and metastatic dissemination via the alteration of the early endosomal routes may be a further development for future cancer studies. To this aim, also STIM1 role as a regulator of cellular homeostasis, via the modulation of endosome positioning and function, such as LE-dependent mTORC1 signaling and autophagy, may be a novel strategy to connect cancer cell endosomal traffic with their metabolism and fate.

## Materials and Methods

### Cell lines and antibodies

Primary venous human endothelial cells (ECs) were isolated from the umbilical cords as previously described (Serini *et al*, 2003). Cells were then cultured in M199 medium completed with cow brain extract, heparin sodium salt from porcine intestinal mucosa (0.025 mg/500 ml), penicillin/streptomycin solution, and 20% fetal bovine serum (FBS; Sigma-Aldrich), in cell culture dishes that had been previously coated with 0.1% gelatin from porcine skin (G9136; Sigma-Aldrich). Cells were tested for mycoplasma contamination by means of Venor GeM Mycoplasma Detection Kit (MP0025-1KT; Sigma-Aldrich). The isolation of primary venous ECs from human umbilical cords was approved by the Office of the General Director and Ethics Committee of the Azienda Sanitaria Ospedaliera Ordine Mauriziano di Torino hospital (protocol approval no. 586, Oct 22 2012 and no. 26884, August 28 2014), and informed consent was obtained from each patient. HEK 293T (ATCC CRL-3216), MRC-5 (ATCC CCL-171), and Hs746T (ATCC HTB-135) cells were grown in DMEM medium completed with glutamine, penicillin/streptomycin

solution, and 10% FBS (Sigma-Aldrich). Both, ECs and HEK 293T, were transfected by means of Lipofectamine and PLUS reagent (Thermo Fisher Scientific).

Mouse monoclonal Abs anti-α-tubulin (B-5-1-2) and anti-KIFC1 (2B9) for both Western blot analysis were from Sigma-Aldrich. Rabbit monoclonal Abs anti-STIM1 ab108994 [EPR3414], anti-Cytoplasmic dynein 1, light-intermediate chain (LIC) ab157468 [EPR11240], anti-p50 (or Dynamitin) ab133492 [EPR5095], anti-HOOK1 ab151756 [EPR10102], anti-KIF5B ab167429 [EPR10276], and monoclonal mouse Ab anti-mCherry ab125096 [1C51] and rat anti-EB1 ab53358 [KT51] and mouse polyclonal Ab anti-HOOK3 ab173388 as well as the rabbit polyclonal anti-FLAG ab1162 were from Abcam and used for Western blot and immunoprecipitation experiments. To immunoprecipitate EB1, mouse monoclonal Ab anti-EB1 (610534) from BD Biosciences was used. Mouse monoclonal anti-p150Glued (610473) used in Western blot and immunoprecipitation and anti-Rab5 (1/Rab5) and anti-LAMP-1 (H4A3) used in immunofluorescence were also from BD Biosciences. Goat polyclonal Ab anti-EEA1 (N-19) for immunofluorescence analysis and the mouse monoclonal anti-Raptor (10E10) used for Western blot were from Santa Cruz Biotechnology. Rabbit polyclonal Ab anti-GFP (A11122) used for Western blot and immunoprecipitation experiments was from Invitrogen. Rabbit monoclonal Ab anti LC3 (1712D) used for Western blot was from Novus Biologicals. Rabbit monoclonal Ab anti V5 (#13202) used in immunoprecipitation or Western blot, and the polyclonal anti-total S6 Kinase 1 (#9202) and anti-phospho S6 Kinase 1 (Thr389, #9205) Abs used for Western blot were from Cell Signaling Technology.

The pH-sensitive fluorescent dye to visualize acidic vesicles was acetoxymethyl (AM) ester-modified fluorogenic intracellular pH probe pHrodo Green AM (Kulkarni et al, 2019) from Thermo Fisher Scientific. Goat anti-rabbit secondary Ab was from Santa Cruz Biotechnology, while goat anti-mouse and rat secondary Abs were from Jackson ImmunoResearch Laboratories. Alexa Fluor 488, 555, and 647 donkey anti-mouse, rabbit, and goat IgG (H + L) secondary Abs were from Invitrogen.

## DNA constructs

pEGFP-*h*STIM1 WT and pEGFP-*h*STIM1 NN (Honnappa et al, 2009) were a kind gift of Anna Akhmanova (Cell Biology Utrecht University, The Netherlands). The pEGFP-*h*STIM1 AA(Yuan et al, 2009) was kindly donated by Shmuel Muallem (National Institute of Dental and Craniofacial Research, NIH, Bethesda, MD, USA).

The coiled coil (CC) deletion mutants, pEGFP-ΔCC1, ΔCC2, ΔCC3, ΔCC1–3/WT, and ΔCC1–3/NN *h*STIM1, were generated by standard PCR protocols according to the Taq polymerase manufacturer's instructions (Fynnzymes) and using pEGFP-STIM1 WT as template. Those deletion constructs were obtained with standard biomolecular techniques. In particular, the mutant pEGFP-ΔCC1 was lacking of the CC1 domain of STIM1 (706–1,023 base pairs; aa 236–341), the mutant pEGFP-ΔCC2 was devoid of the CC2 domain of STIM1 (1,084–1,161 base pairs; aa 362–387), and the pEGFP-ΔCC3 was lacking of the CC3 domain of STIM1 (1,192–1,263 base pairs; aa 398–421). Furthermore, the ΔCC/WT and ΔCC/NN deletion mutants were removed of the all three CC domains of STIM1 (706–1,263 base pairs; aa 236–421; ≅ 20 kDa) in their respective pEGFP-STIM1 WT or NN backbones.

The cDNA of mCherry-Dynactin-N-18 was a gift from Michael Davidson (Addgene plasmid # 55034; http://n2t.net/addgene:55034; RRID: Addgene_55034).

## Purified protein production and pulldown experiments

The cDNA and protein of p150Glued N-terminal Cap-Gly, with its first coiled coil (CC) domain (NCBI Reference Sequence: NP_004073.2, AA 1–358), C-terminally tagged with V5 (GKPIPN PLLGLDST) were generated (ordered gene ID #391071) and purchased from ATUM Bio (Newark, CA). The same was done with the C-terminal portion (ordered gene ID #394359) of EB1 protein WT (NCBI Reference Sequence: NP_036457.1, AA 191–268), N-terminally tagged with FLAG (DYKDDDDK) and its deletion mutant form (EB1-ΔY, AA 191–267), from which the tyrosine residue of the final EEY motif was deleted (ordered gene ID #394360), as previously reported (Komarova et al, 2005). The cDNA of the cytoplasmic domain of STIM1 (NCBI Reference Sequence: NP_001264890.1, AA 235–791), N-terminally tagged with Glutathione S-Transferase (GST), was kindly donated by Francisco Javier Martin-Romero (University of Extremadura, Badajoz, Spain). GST-cytoplasmic STIM1 and empty GST (pGEX-3X vector), used as negative control in the pulldown assays, were produced and purified. Briefly, BL21 transformants were plated on LB agar with 50 mg/l ampicillin or 30 mg/l kanamycin and incubated overnight at 37°C. One colony from each transformation was picked and grown into 50 ml of TB medium containing 50 mg/l ampicillin or 30 mg/l kanamycin and then incubated overnight at 37°C. Overnight culture was inoculated into 1 l of TB medium containing antibiotics and incubated at 37°C until an OD 600 of 0.8 was reached and then induced with 1 mM IPTG for 3 h. Cells were harvested by centrifugation and ice-cold lysed in 20 mM Tris–HCl at pH 7.4, supplemented with DNAse I (Roche) and complete protease inhibitors (Roche). Total and soluble protein fractions were denatured and run on polyacrylamide gel under reduced conditions. Expression levels were estimated by densitometry. P150Glued and EB1 fragments in clarified lysate were purified using streptavidin-column based methods by ATUM Bio, whereas GST alone or bound to STIM1 were purified by incubating lysate with Glutathione Sepharose 4B resin (GE Healthcare) for 2 h at 4°C. All proteins were eluted in 20 mM Tris–HCl at pH 7.4, 350 mM NaCl and 1mM 2-mercaptoethanol, as previously described by Thomas Surrey and collaborators (Duellberg et al, 2014).

For pulldown assays, empty or cytoplasmic-STIM1-bound GST proteins were combined in an 1:1 ratio with dissolved purified p150Glued-V5-tagged fragment, in the presence or absence of FLAG-tagged EB1 WT or ΔY, in a buffer containing 20 mM Tris–HCl at pH 7.4, 350 mM NaCl, and 1 mM 2-mercaptoethanol (Duellberg et al, 2014). After incubation for 1 h on a rotating wheel at 4°C, the protein mixture was added on anti-V5 rabbit monoclonal antibody pre-coated beads, for 2 h on a rotating wheel at 4°C. Beads were then separated from the supernatant by centrifugation and washed four times in the same buffer, and the proteins retained on the beads were analyzed by Western blotting.

## Immunoprecipitation and Western blot analysis

To co-immunoprecipitate and analyze by Western blotting STIM1 constructs and the dynein–dynactin complex components, ECs or

HEK293Ts were lysed in buffer containing 25 mM Tris–HCl pH 7.2, 150 mM NaCl, 1% NP-40, 5% glycerol, 5 mM $MgCl_2$, 1 mM PMSF, 1 mM $Na_3VO_4$, and protease inhibitor cocktail.

Cellular lysates were incubated for 20 min on wet ice and then centrifuged at 15,000 g, 20 min, at 4°C. The total protein amount was determined using the bicinchoninic acid (BCA) assay (Pierce). Equivalent amounts (1 mg) of protein were immunoprecipitated for 2 h at 4°C with the antibody of interest, and immune complexes were recovered on protein A- or G-Sepharose (GE Healthcare) for 1 h at 4°C. Where indicated in the figure legend, cell lysates were pre-cleared on Sepharose beads before incubation with the antibody of interest. Controls of immunoprecipitations were performed incubating cell lysate with empty immunoglobulin (Ig) G of the same specie of the antibody of interest (Normal rabbit, NI#01, or mouse, NI#03, IgG from Merck Life Science). Immunoprecipitates were washed four times with lysis buffer with or without detergent and then separated by SDS–PAGE. Proteins were then transferred to a Hybond-C extra nitrocellulose membrane (Amersham), probed with antibodies of interest, and detected by enhanced chemiluminescence technique (PerkinElmer).

**Gene silencing and mRNA Real-Time PCR quantification**

For siRNA-mediated silencing, the day before oligofection, ECs were seeded in six-well dishes at a concentration of $10 \times 10^4$ cells/well. Oligofection of siRNA duplexes was performed according to manufacturer's protocols. Briefly, human ECs were transfected twice (at 0 and 24 h) with 200 pmol of siGENOME Non-Targeting siRNA Pool #1 (D-001206-13) as control (siCTL) or siGENOME SMART pools (Dharmacon) for human STIM1 (M-011785-00), KIFC1 (M-004958-02) or KIF5B (M-008867-00) were used. The two siRNAs against STIM1 (siSTIM1) and KIFC1 (siKIFC1) were also given to ECs together (400 pmol total). Forty-eight h after the second oligofection, ECs were lysed or tested in functional assays. To silence human HOOK1 or HOOK3, 200 pmol of IDT Non-Targeting DsiRNA (#51-01-14-04) as control (siCTL), IDT hs.Ri.HOOK1.13.3 DsiRNA (#76-66-51-25) or IDT hs.Ri.HOOK3.13.3 DsiRNA (#77-07-02-20) were used, singularly and simultaneously as described.

Lentiviral vectors carrying short hairpin RNAs (shRNA) sequences against STIM1 (TRCN0000179490, #490 in Fig EV1A; TRC0000358718, #718 in Fig EV1A; TRC0000358780, #780 in Fig EV1A) were from the RNAi Consortium library (Sigma-Aldrich). Only the shSTIM1 #780 was used in the experiments shown in this manuscript. Control cells (shCTL) were transduced with a lentiviral preparation carrying the empty pLKO vector of the same RNAi Consortium library.

KIFC1 (Hs00382558_m1), KIFC2 (Hs01057295_g1), and KIFC3 (Hs00194304_m1) mRNA expression levels were measured by Real-Time PCR using Taqman Gene Expression Assays (Thermo Fisher).

**Confocal microscopy and early/late endosome size, number, and distance to nucleus quantification**

Control or silenced EC cells were plated on glass coverslips coated with 0.1% gelatin from porcine skin (G9136, Sigma-Aldrich) and allowed to adhere overnight in a 24-well plate. Cells were washed in phosphate-buffered saline (PBS), fixed in 4% paraformaldehyde

(PFA), permeabilized in 0.01% saponin for 5 min on ice, incubated with different primary Abs for 1 h, and revealed by appropriate Alexa Fluor-tagged secondary Ab (Molecular Probes by Life Technologies). Cells were analyzed by using a Leica TCS SP8 AOBS confocal microscope equipped with two hybrid detectors (HyD) that, by combining classical photomultipliers with highly sensitive avalanche photodiodes, provides higher signal-to-noise ratio, image contrast, and sensitivity. PL APO 100×/1.4 NA immersion objective was employed. 1,024 × 1,024 pixel images were acquired at pixel size = 87.30–92.26 nm, and a z-stack of 1.49 μm (spanned over steps ≤ 0.37 μm) was acquired. For nucleus staining in immunofluorescence images taken with the Leica TCS SP8, DRAQ5 Fluorescent Probe Solution (Thermo Fisher) was used. Immunofluorescence analysis was performed as previously described (Mana et al, 2016). Colocalization analysis to calculate Pearson correlation was performed with the Leica Confocal Software Quantification Tool (Leica Microsystems). Image acquisition was performed by adopting a laser power, gain, and offset settings that allowed maintaining pixel intensities (gray scale) within the 0–255 range and hence avoid saturation.

In some immunofluorescence experiments, ECs, MRC-5s, or Hs746Ts were plated on glass 0.1% gelatin from porcine skin (G9136, Sigma-Aldrich)-coated coverslips in a 24-well plate, and the day after, they were treated with 120 μM Ciliobrevin D (CilioD; EMD Millipore) for 1 h or with 5 μM AZ82 (Aobious) for 5 min or with 4 μM Thapsigargin (TG; Sigma-Aldrich) for 5 min and diluted in cell complete medium at 37°C, 5% $CO_2$ in a humidified atmosphere. For double treated cells, after 55 min of Ciliobrevin D treatment, AZ82 was also added to the medium for the lasting 5 min. Afterward, cells were gently washed and fixed in 4% PFA.

MATLAB software codes were developed in order to automatically quantify (i) nucleus position, (ii) cell size, and (iii) the size, number, and distance to nucleus of EEs and LEs. The custom-made code exploits MATLAB built-in functions to segment maximum intensity projection images resulting in image segmentation like those shown in Fig EV3E.

**Identification of nuclei**

The algorithm analyses one cell at a time, and it first identifies the nucleus by isolating only the blue channel and segmenting such image through a thresholding method. This thresholding method is kept as starting step for the detection of nuclei, cells, and endosomes, and it can be summarized as follows. Once a threshold value is determined, the original image is converted into a binary image by automatically setting to 1 (white) all those pixels overcoming the threshold and to 0 (black) all the other pixels. Then, each single object composed of eight-connected white pixels is labeled. The threshold value is set as the average of the positive pixels of the image plus $n$ times the standard deviation, with the value of $n$ adjusted depending on the fluorescent intensity of the image. Spatially near-white pixels are then connected in order to form a single object (Fig EV3E, Nucleus), which is subsequently re-labeled. At this stage, if more than one object is found, only the biggest one is saved and considered as the nucleus. The spatial position of both the entire nucleus and its centroid (white dot in Fig EV3E) is computed. The former is used to discard the area corresponding to the nucleus during the endosome segmentation procedure, while the latter is used to compute the endosomes distance to the nucleus.

### Identification of cells

To analyze cell size, we considered the average of fluorescent signals coming from both EE (green) and LE (red) channels. Images (Fig EV3E, Cell green/red channel) are first segmented through the thresholding method described above (with the threshold set to zero to maximize the area covered by the cell), and then, image manipulations are performed in order to obtain a single and solid object. Finally, the conversion in micrometers of the total number of pixels composing the average segmented area gives the cell area.

### Identification of EEs and LEs

Through the thresholding method described above, the EEA-1 or LAMP-1 staining image is segmented to identify single EE or LE objects with high level of fluorescence. Subsequent morphology-based functions are applied to separate merged objects. To avoid artifacts, all those objects whose size is lower than a threshold ($s\_min$) are discarded. The threshold $s\_min$ has been set to three pixels (corresponding to object diameter ≤ 150 nm) for EEs, while it is, depending on the experiment, 10 or 20 pixels (corresponding to object diameter ≤ 310 nm or ≤ 440 nm, respectively) for LEs. If some objects, bigger than the respective $s\_min$, are segmented as ring-like shape objects, are then filled. The segmented objects are finally labeled and represent the EE or LE vesicle population of the analyzed cell (Fig EV3E, EE and LE). Their size and centroid position (used for the distance to nucleus computation) are then measured through MATLAB built-in functions.

For large and clustered vesicles such as LEs, some objects are segmented as grape-like shape structures, formed by differently sized ring-like shape objects. If the hole's ring size is smaller than $s\_min$, that object is discarded and its area is included in the grape shape object, otherwise it is converted into a new LE, whose pixel value is set to 1 and the rest of the object is discarded (pixel value set to 0). Such procedure is repeated for each object of each grape-like structure, and objects are re-labeled. For the hole's ring analysis, the Euler number, as the difference between the number of grape-like structures and the number of holes' rings in those structures, has been calculated for each re-labeled object. Afterward, an additional check level is performed. Indeed, a circularity test has been performed and only the objects with circularity values ($c$) within the range of 0.6 and 1.5 are kept. To this purpose, we calculated $c$ as the ratio between the square of each object perimeter and four times its area, being $c = 1$ in a perfectly circular object. The remaining objects are the final LE population, whose total number, size, and distance to nucleus are computed.

Given the absence of cell size bias introduced by some silencing and treatment experiments (Fig EV3F), the normalized nucleus distance (as that between the nucleus and EE or LE centroid) of each vesicle is then plotted. For conditions, significantly influencing cell size, the distribution of not normalized distance to nucleus is shown for clarity together with the cellular diameter on the corresponding plot.

### Live imaging experiments and overtime early endosomes positioning and retrograde motion quantification

EC cells, plated in a six-well glass bottom, 0.1% gelatin from porcine skin (G9136, Sigma-Aldrich) -coated, black-sided plate (Cellvis), were transiently transfected with the desired fluorescent-tagged cDNAs or oligofected with STIM1 siRNA and allowed to adhere overnight. To visualize Rab5+ early endosomes (EEs), CellLight Early-GFP BacMam 2.0 (Thermo Fisher) was used. Cells were analyzed by using a Leica TCS SP8 AOBS confocal microscope equipped with two HyD, as described in the previous Confocal microscopy method section (HyD). PL APO 63×/1.4 NA immersion objective was employed. 512 × 512 pixel images were acquired at pixel size = 161.51–182.01 nm. Confocal image thickness along $z$-axis was kept constant at 0.9 μm throughout the experiments. This value results to be particularly relevant using endothelial cells, being flat and wide cells, whose thickness is about 1 μm in the area away from the nucleus. Image acquisition was performed by adopting a laser power, gain, and offset settings that allowed maintaining pixel intensities (gray scale) within the 0–255 range and hence avoids saturation. Movies to quantify the different physical parameters of Rab5+ EEs were acquired for 2 min, taking one frame every 0.5 s.

To automatically measure the distance to the nucleus of Rab5+ EEs, an image segmentation and particle tracking algorithm was developed by exploiting MATLAB built-in functions. The software analyses one movie at a time and through the thresholding segmentation method described in the previous Confocal microscopy method section, the endosomes are recognized in each frame and their centroid position is computed. The threshold is set as the average of the image pixels' intensities, plus $n$ times their standard deviation ($n$ is adjusted depending on the experiment). In order to avoid possible artifacts, we considered as endosomes only those segmented objects composed by at least three pixels (object size ≥ 300 nm). The tracking of endosomes over time is performed by checking frame by frame whether the position of each single recognized endosome matches with any of the endosomes of the previous frame within a range of five pixels. If more than one match is encountered, the closest to the one of the previous frame is accepted. If no matches are found, the endosome is considered as "disappeared". Through such method, when an endosome disappears and another one appears in its same position after two frames, the second one is considered a "new endosome". For the distance to the nucleus in STIM1 silencing experiments, only those EEs appearing consecutively for 60 s are accounted, whereas in the treatment experiments only those appearing from 30 s are considered. The nucleus position in the frame is measured by considering its centroid. Given the absence of cell size bias introduced by STIM1 silencing and CilioD or AZ82 treatments (Fig EV3F), the normalized to the nucleus distance (as that between the nucleus and EE centroid) of each vesicle is then plotted. The distance to the nucleus of Rab5+ EEs was also measured in EC cells treated with Ciliobrevin D, AZ82, or a combination of the two drugs, filming one frame every 0.5 s. For double CilioD + AZ82 treatment, cells were given the dynein inhibitor for 55 min, and then, AZ82 was added to the medium. To distinguish centripetal-directed Rab5+ EEs and track their retrograde velocity, wild-type EC cells, transduced 24 h before with CellLight Early-GFP BacMam 2.0, were treated with CilioD or AZ82, as previously described, and filmed (one frame every 0.5 s). To automatically compute the retrograde velocity of EEs, among all those EE appearing for 30 s, we firstly selected all those approaching the nucleus and then calculated their velocity. To determine the direction of motion, for each endosome, we performed a linear fit of its distance to the nucleus over time. A positive slope (average increasing distance to the nucleus overtime) is a proxy for a

centrifugal motion, while a negative slope (average decreasing distance to the nucleus) is a proxy for a centripetal, or retrograde, motion. To our purposes, only the latter ones are selected and their velocity is computed by dividing the effective path by the duration of the trajectory (30 s). The data were analyzed with MATLAB software (MATLAB R2016b).

### Statistical analysis

For statistical evaluation, parametric two-tailed heteroscedastic Student's *t*-test was used to assess the statistical significance when two groups of unpaired normally distributed values where compared; when more than two groups were compared, parametric two-tailed analysis of variance (ANOVA) with Bonferroni *post hoc* correction was applied. For mRNA and Western blot analysis of cell lysates and immunoprecipitations, standard deviation (SD) is shown. For all other quantifications, standard error of the mean (SEM) is shown. For all distance to the nucleus quantifications, the probability of vesicle positioning respect to the nucleus is represented in a distribution plot. For LE siCTL-siSTIM1 size comparison (Fig 2B), the two-sample Kolmogorov–Smirnov test (KS test) was used with MATLAB software, testing the alternative hypothesis that the siCTL LE distribution had longer tail than the siSTIM1 LE one. Statistical differences were considered not significant (ns) = $P$-value > 0.05; significant * = $P$-value $\leq$ 0.05; ** = $P$-value $\leq$ 0.01; *** = $P$-value $\leq$ 0.001).

## Data availability

This study includes no data deposited in external repositories.

**Expanded View** for this article is available online.

## Acknowledgements
Supported by Fondazione AIRC (IG grants #16702 and 21315 to G.S.); Fondazione AIRC under 5 per Mille 2018 - ID. 21052 program – P.I. Comoglio Paolo, G.L. Serini Guido (to G.S.); FPRC-ONLUS Grant "FPRC - 5 per mille 2014 Ministero Salute" (to G.S.) and "5 per mille MIUR 2012 -FPRC Innovation Grant" (to G.V.); Associazione "Augusto per la Vita" (to G.S.); Fondazione Telethon (grant n. GGP15102) (to G.S.); G.V. was supported by a fellowship from FPRC-ONLUS (5 per mille MIUR 2012 -FPRC Innovation Grant).

## Author contributions
GS and GV conceived the project. GS, GV, CEB, MDG, CS, NG, CC, AFP, and CB designed the experiments. GV, CEB, MDG, CS, NG, and CC performed the experiments. GS, GV, CEB, MDG, CS, NG, CC, AFP, and CB analyzed the data. GS, GV, CEB, MDG, CS, NG, CC, AFP, and CB interpreted the results. GS, GV, CEB, and CB wrote the paper. All authors read and approved the manuscript.

## Conflict of interest
The authors declare that they have no conflict of interest.

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
