## [Review Process File · The EMBO Journal]

Distinct retrograde microtubule motor sets drive early and late endosome transport

Guido Serini, Giulia Villari, Chiara Enrico Bena, Marco Del Giudice, Noemi Gioelli, Chiara Sandri, Chiara Camillo, Alessandra Fiorio Pla, and Carla Bosia

DOI: [10.15252/embj.20193661](https://doi.org/10.15252/embj.20193661)

Corresponding author: Guido Serini (guido.serini@ircc.it)

Review Timeline:

Submission Date:	9th Oct 19
Editorial Decision:	14th Nov 19
Authors' Appeal:	23rd Nov 19
Editorial Decision:	3rd Dec 19
Revision Received:	26th Jun 20
Editorial Decision:	4th Aug 20
Revision Received:	1st Oct 20
Accepted:	14th Oct 20

Editor: Ieva Gailite

Transaction Report:

Thank you for submitting your manuscript to The EMBO Journal. We have now received three referee reports on your manuscript, which are included below for your information. Based on these comments, we unfortunately had to conclude that the study is not a sufficiently strong candidate for publication in The EMBO Journal.

As you will see, while all reviewers find the topic of interest, they also raise numerous substantive concerns regarding the experimental setup, data analysis and interpretation, and they find that the main conclusions of the manuscript are not sufficiently supported by the current data. Given these opinions from good experts in the field, I am afraid that we cannot offer further proceedings towards publication in The EMBO Journal.

Thank you in any case for the opportunity to consider this manuscript. I regret that I cannot communicate more positive news, but I nevertheless hope that you will find the comments of our reviewers helpful.

Referee #1:

Villari et al. report that the differential localizations of early and late endosomes are mediated by the specific involvement of the minus end-directed motor KIFC1 on early endosomes. The authors report that STIM1 interacts with the microtubule plus end binding protein EB1 via its SxIP motif, and this promotes its association with the dynactin/dynein subunit p150Glued. The authors found that the peripheral distribution of early endosomes requires a simultaneous interaction of early endosomes (via HOOK1/3) with dynein and KIFC1. Surprisingly, the authors found that siRNA-mediated depletion of STIM1 led to a peripheral redistribution of LAMP1-positive late endosomes whereas Rab5-positive early endosomes redistributed to the perinuclear area. Even though this manuscript describes several novel molecular interactions, it fails to provide a convincing mechanistic explanation for the differential localization of early and late endosomes. The authors have only studied minus-end-directed microtubule motors, but it is well known that plus-end-directed motors play a major role in early and late endosome positioning. The failure to include such motors in the analyses preclude the authors from drawing firm conclusions about the mechanisms that differentiate between early and late endosome positioning.

Major points:

1. The present data do not provide a plausible explanation for the differential localization of early and late endosomes, especially since both dynein and KIF1C are minus-end-directed microtubule

motors. The authors need to study whether plus-end-directed motors such as KIF2 and KIF5 also associate preferentially with one of the two endosome populations. If so, then endosome positioning could equally well be explained by alternative mechanisms.

2. The functional implications of the reported effects on endosome distributions remain unclear. The authors need to test effects on biological activities that have been reported to depend on endosome positioning, such as cargo degradation, recycling and mTORC1 signaling.

3. The authors should consider the possibility that interference with motor functions might affect endosome maturation. Using EEA1 as an early endosome marker and LAMP1 as a late endosome marker might be confounded if motor depletion affects endosome maturation.

Minor point:

Bar diagrams should be replaced by scatter plots, which give a visualization of data distribution.

Referee #2:

The mechanisms of transport of organelles is an exciting field and critical for the understanding of cell biology or better... the biology of the cell. Here the authors propose that late and early endosomes differ with respect to the molecular mechanism of retrograde transport. Late endosomes solely depend on dynein:p150glued for minus-end MT transport while retrograde early endosome transport occurs either via Dynein:p150glued:HOOK1 or KIF1C:p150glued:HOOK3. Why early endosomes utilizes two retrograde transport systems remains unclear. The experiments depend on microscopy analyses of endosomes in cells silenced for the different proteins, sometimes overexpression of fragment of these proteins and cIP experiments followed by WB for the analyses of interactions to proteins or their domains. They are not always very convincing. For example, Fig 1 shows the interaction between STIM1 and p150 by cIP/WB and by confocal. Fig 1b shows a very poor cIP between overexpressed STIM1 and p150 that is 1 band. Fig 1c shows the same interaction with STIM1 fragments but now reveals 2(!) p150 bands. Also the microscopy in Fig 1e shows that p150 decorated microtubules and two selected dots with some colabelling (not that the blob is different). However, magnification of the figures shows that the colabelling is in fact very poor. It would also be helpful if the authors test the interaction with purified domains. In figure 2 the authors quantify the late endosome size, which is remarkable given the quality of the pictures. How the authors arrive at the extremely small SEM in their quantification, is surprising, especially when measuring such small endosomes. However, a more substantial problem (and that is throughout this manuscript) is that STIM1 is only silenced with one shRNA or a pool of siRNA and off target effects can then not be excluded. This is not the standard in cell biology tmo.

In conclusion, the hypothesis is of interest and the topic is definitively of interest. But the quality of the data is insufficient to substantiate the claims in this manuscripts.

Other points.

- A cell is a 3D system. However, the authors only show data from 2D images. Endosomes are typically clustered in the perinuclear area, however the image plane will largely affect the number of

imaged endosomes as well as their localization (peripheral endosomes are mainly found closer to the glass surface). The authors should consider analysing the whole cell (3D) or at least indicate that the image plane is constant/similar between all conditions shown. No cell boundaries are indicated in any of the pictures (or I might fail to see it on the print/screen).

- The IP controls are rather unusual. 1. They incubate lysates with empty beads as control for endogenous immunoprecipitation. If the authors would like to use such controls they should at least coat the beads with isotype controls/serum. Better would be to just compare the control with the depleted situation (which they also did occasionally). 2. Their overexpression WBs lack proper washing controls. They incubate lysates (for example GFP/RFP transfected) with empty beads. They should include the Ab used for immunoprecipitation also in this sample to show that the co-immunoprecipitated proteins does NOT bind under EV conditions. Their WBs are anyhow confusing since in 1C there is a band in the ctrl while there is no GFP-STIM1 present In addition, Fig2C: How come the IPed band for p150glued runs higher than p150glued in the TL? Also the position of protein marker standards should be shown
- The authors show (most of the time) images with only one phenotypic cell. Seeding the cells a bit more dense will allow them to show multiple cells at ones which will strengthen their conclusions.
- The authors exclude clustered endosomes for their analyses. Is it fair to do this? In control situation the majority of LEs are clustered, are these all excluded? For example in Fig2B they analysed a total of 150 endosomes from 30 cells. That means 5 endosomes per cell? I guess I misunderstood this, but then correct. Also STIM1 depletion leads to more dispersed endosomes which the authors conclude are smaller in size compared to endosomes in the control. Endosomes are known to be smaller in the periphery/ or enlarge a bit due to fusion events if they reach the PN area (as the authors also point out). If they exclude all the clustered endo's (most likely bigger ones), they may automatically affect endosomal size, which would then not be an informative result.
- It is misleading to use SEM for (WB) quantifications., this results in super small and unrealistic error bars. The authors should use SD instead and preferably plot single values.
- Remaining questions: what determines if Dynein or KIFC1 is in charge of EE transport? Are Hook3 and Hook 1 on similar or distinct EE vesicles? This part of the paper is very limited while this part actually contains their main and most interesting message.

Other comments:

- To deplete cells for STIM1 they use a pool of siRNAs, but they see a similar phenotype using shRNA (so they do use multiple).

Referee #3:

In this manuscript, Villari et al report an interaction between the endoplasmic reticulum (ER) transmembrane protein stromal interaction molecule 1 (STIM1) and p150Glued, a subunit of the dynactin complex. They show that STIM1 promotes the localization of p150Glued at the microtubule (MT) plus ends through the formation of a ternary complex with the MT end-binding protein 1 (EB1). The authors then investigate the role of STIM1 in the distribution of early (EEs) and late endosomes/lysosomes (LEs). Silencing of STIM1 leads to opposite effects, LEs being redistributed toward the cell periphery whereas peripheral EEs are clustered around the nucleus. In a second part of the manuscript, the authors further explore the mechanisms responsible for the peripheral localization of EEs. They provide evidence that their localization relies on the antagonistic action between dynein and the MT minus-end motor KIFC1, which are attached to EEs via HOOK1 and HOOK3 adaptor complexes, respectively.

STIM1 is an ER Ca²⁺ sensor that plays a key role in store-operated calcium entry (SOCE) by interacting with the plasma membrane (PM) Ca²⁺ channel Orai1 following depletion of ER Ca²⁺ stores. STIM1 interaction with EB1 is thought to regulate its translocation to PM-ER contact sites. That STIM1 is also involved in the distribution and transport of endosomal compartments is a novel finding of potential interest. Nevertheless, the manuscript suffers from major weaknesses and in its present form, does not meet the necessary standards for a publication in EMBO J.

Major comments:

1) Figure 1: The authors claim that STIM1/p150Glued/EB1 forms a ternary complex. This is not directly proven, Figs 1A-C simply showing protein interactions two by two. In addition, experiments were performed following overexpression of tagged STIM1 and p150Glued. Did the authors try to co-IP endogenous proteins?

The results presented in Fig 1C and 1G seem contradictory. In Fig.1C, the NN STIM1 mutant (unable to bind EB1) can be co-IP with p150Glued but does not co-localize with it by IF (Fig. 1G).
2) Figure 2: The authors conclude for a dual role of STIM1 in SOCE and LE distribution based on the rescued effect of the AA STIM1 mutant (unable to bind Orai1) (Figs 2E, F). This is not enough. The authors should investigate the effect of the depletion of ER calcium stores on the distribution of LEs. These experiments are also important to rule out the possibility that the effects of STIM1 depletion on LE distribution result from changes in ER morphology.

3) Figures 5 and 6: The localization of KIFC1 should be provided. Based on the effects of silencing both KIFC1 and STIM1 or inhibiting KIF1C and dynein, the authors suggest that KIFC1 and STIM1/dynein "antagonistically" cooperate. KIFC1 and dynein/dynactin complex being both MT-end motors, how can it work? No mechanism(s) is (are) proposed.

4) Figure 7: IP experiments shown in Fig. 7A-C are incomplete. Does HOOK3 antibody coimmunoprecipitate KIFC1? Why KIFC1 was not probed in Fig. 1A? Fig. 7E should be quantified.

5) Figure S3C: The hypothesis that STIM1/dynein compete for binding to p150Glued is interesting but should be supported by stronger results. Does KIFC1 silencing affect binding of p150Glued to HOOK1?

6) It is written in almost all Fig. legends that quantification of endosomes in fixed cells were performed "for a total of 150 endosomes (30 cells)". Does it mean that only 5 endosomes have been analyzed per cell?

Other comments:

7) The contribution of centrosomal MTs to EE localization should be at least discussed. A + end kinesin motor, KIF16B, was shown to modulate the distribution of EEs (Hoepfner et al, Cell 2005).

8) Figure 1C: Why does the GFP-STIM CC mutant (deletion of the 3 coiled coil domains) show exactly the same electrophoretic motility than WT? In Fig. S1, delta CC2 and CC3 mutants migrate faster than WT.

9) The authors claim that they verify "the ability of ER protein STIM1 to bind, similarly to p150Glued, EB1 (Grigoriev et al. 2008). The STIM1-p150Glued interaction has not been documented in that manuscript.

EMBOJ-2019-103661

Villari et al., Distinct retrograde motor sets drive early and late endosome transport

Referee #1:

Villari et al. report that the differential localizations of early and late endosomes are mediated by the specific involvement of the minus end-directed motor KIFC1 on early endosomes. The authors report that STIM1 interacts with the microtubule plus end binding protein EB1 via its SxIP motif, and this promotes its association with the dynactin/dynein subunit p150Glued. The authors found that the peripheral distribution of early endosomes requires a simultaneous interaction of early endosomes (via HOOK1/3) with dynein and KIFC1. Surprisingly, the authors found that siRNA-mediated depletion of STIM1 led to a peripheral redistribution of LAMP1-positive late endosomes whereas Rab5-positive early endosomes redistributed to the perinuclear area.

Even though this manuscript describes several novel molecular interactions, it fails to provide a convincing mechanistic explanation for the differential localization of early and late endosomes. The authors have only studied minus-end-directed microtubule motors, but it is well known that plus-end-directed motors play a major role in early and late endosome positioning. The failure to include such motors in the analyses preclude the authors from drawing firm conclusions about the mechanisms that differentiate between early and late endosome positioning.

Major points:

1. The present data do not provide a plausible explanation for the differential localization of early and late endosomes, especially since both dynein and KIF1C are minus-end-directed microtubule motors. The authors need to study whether plus-end-directed motors such as KIF2 and KIF5 also associate preferentially with one of the two endosome populations. If so, then endosome positioning could equally well be explained by alternative mechanisms.

This point is definitely relevant for a more complete understanding of the mechanism, so we will silence both KIF2 and KIF5 singularly and in combination with both dynein and KIFC1 inhibitors to check early endosome positioning.

2. The functional implications of the reported effects on endosome distributions remain unclear. The authors need to test effects on biological activities that have been reported to depend on endosome positioning, such as cargo degradation, recycling and mTORC1 signaling.

We did test cargo (i.e. β 1 integrin) recycling after STIM1 silencing and we observed a significant increase of it in EEA1⁺ endosomes in both a biochemical assay (ELISA) and localization study by immunofluorescence. We didn't include those data in the present version of the manuscript, but we will definitely do it in the next submission. Moreover, we will check whether STIM1 silencing also affects mTORC1 signaling.

3. The authors should consider the possibility that interference with motor functions might affect endosome maturation. Using EEA1 as an early endosome marker and LAMP1 as a late endosome marker might be confounded if motor depletion affects endosome maturation.

We are aware that the interference with motor function may also affect endosome maturation. However, in Figure 3A we already do show that the observed phenotype after STIM1 silencing is also seen using Rab5 as a marker of early endosomes, so

the two markers (Rab5 and its effector EEA1) are comparable. At present, we could not repeat the same staining for Rab7 as we did not find a good Rab7 antibody among the many we tested so far.

Minor point:

Bar diagrams should be replaced by scatter plots, which give a visualization of data distribution.

We totally agree, **we will change the plots.**

Referee #2:

The mechanisms of transport of organelles is an exciting field and critical for the understanding of cell biology or better... the biology of the cell. Here the authors propose that late and early endosomes differ with respect to the molecular mechanism of retrograde transport. Late endosomes solely depend on dynein:p150glued for minus-end MT transport while retrograde early endosome transport occurs either via Dynein:p150glued:HOOK1 or KIF1C:p150glued:HOOK3. Why early endosomes utilizes two retrograde transport systems remains unclear.

- The experiments depend on microscopy analyses of endosomes in cells silenced for the different proteins, sometimes overexpression of fragment of these proteins and cIP experiments followed by WB for the analyses of interactions to proteins or their domains. They are not always very convincing. For example, Fig 1 shows the interaction between STIM1 and p150 by cIP/WB and by confocal. Fig 1b shows a very poor cIP between overexpressed STIM1 and p150 that is 1 band. Fig 1c shows the same interaction with STIM1 fragments but now reveals 2(!) p150 bands.

This statement of the reviewer is not correct. Indeed, Figure 1b shows endogenous p150Glued coming down in colIP with endogenous STIM1, whereas Figure 1c only shows overexpressed mCherry-p150Glued colIP together with GFP-tagged STIM1 mutants. Although we agree with the reviewer and we will then change the blot in c, the two bands came up because (as specified in panel c) the p150Glued antibody detected both the endogenous (lower band) and the mCherry-tagged version (upper band) of the protein.

- Also the microscopy in Fig 1e shows that p150 decorated microtubules and two selected dots with some colabelling (not that the blob is different). However, magnification of the figures shows that the co-labelling is in fact very poor.

In agreement with what previously described by the Akhmanova lab (Grigoriev et al., 2008, Curr Biol 18: 177-82), our images show that WT STIM1 is enriched in punctate structures localized at MT plus ends, where proteins such as EB1 and p150Glued are enriched. Is reviewer referring to STIM1 WT-decorated MT plus ends when s/he speaks about 'blobs'? We agree with the reviewer that the STIM1-WT/p150Glued co-labeling along MT is poor, yet **this statement of the reviewer is not correct.** Indeed, as also clearly specified in graphs, we quantified the co-localization at MT plus-ends, not along the whole MT length.

- It would also be helpful if the authors test the interaction with purified domains.

We are planning **domain protein production and interaction analyses.**

- In figure 2 the authors quantify the late endosome size, which is remarkable given the quality of the pictures.

We assume the reviewer is referring to **magnification of the pictures** we decided to show. **Showing a representative whole cell is the only way to illustrate the defective distribution of late endosomes caused by STIM1 silencing.** Images were acquired Leica TCS SP2 or SP8 AOBS confocal laser-scanning microscopes (Leica Microsystems, Wetzlar, Germany). PL APO 63 x/1.4 NA immersion objectives was employed. **1024x1024 pixel images were acquire at pixel size = 87,30-92,26 nm.** Hence, **we have much more than enough resolution to additionally show close ups** in which the different size of endosomes may be better appreciated, while losing the information about the global distribution of endosomes, which is the main focus of the manuscript.

- How the authors arrive at the extremely small SEM in their quantification, is surprising, especially when measuring such small endosomes.

Raw data, which, as per lab (and EMBO J.) policy, we collected and organized in a dedicated Excel file, will be provided at the submission. So that, direct evaluation and re-quantification by others will be possible already during the review process. Furthermore, **to increase the endosome number counted per cell, we will extend the automated method, used to quantify early endosome parameters in live imaging experiments as in Fig 6B,** also to fixed staining of both late and early endosomes in rescued experiments in Figures 2, 3, 4, 5, and 7.

- However, a more substantial problem (and that is throughout this manuscript) is that STIM1 is only silenced with one shRNA or a pool of siRNA and off target effects can then not be excluded. This is not the standard in cell biology tmo.

This statement is not correct and it is contradicting another statement of the same reviewer. Indeed, as highlighted below by the reviewer her/himself (in the “other comments” section of his/her report), we performed our experiments with a pool of siRNAs and a shRNA and we saw a similar phenotype, so we do use multiple.

In conclusion, the hypothesis is of interest and the topic is definitively of interest. But the quality of the data is insufficient to substantiate the claims in this manuscripts.

Other points.

- A cell is a 3D system. However, the authors only show data from 2D images. Endosomes are typically clustered in the perinuclear area, however the image plane will largely affect the number of imaged endosomes as well as their localization (peripheral endosomes are mainly found closer to the glass surface). The authors should consider analysing the whole cell (3D) or at least indicate that the image plane is constant/similar between all conditions shown. No cell boundaries are indicated in any of the pictures (or I might fail to see it on the print/screen).

The confocal image z plane was constant between all conditions shown, **being 0,9 µm,** something which is **particularly relevant since endothelial cell thickness** is about 2 µm in the nuclear area and 1 µm in the area away from the nucleus.

We will indicate cell boundaries in pictures.

- The IP controls are rather unusual.
 1. They incubate lysates with empty beads as control for endogenous immunoprecipitation. If the authors would like to use such controls they should at least coat the beads with isotype controls/serum. Better would be to just compare the control with the depleted situation (which they also did occasionally).

We will re-perform the immunoprecipitation experiments with beads coated with control antibodies where appropriate, as requested.

2. Their overexpression WBs lack proper washing controls. They incubate lysates (for example GFP/RFP transfected) with empty beads. They should include the Ab used for immunoprecipitation also in this sample to show that the co-immunoprecipitated proteins does NOT bind under EV conditions. Their WBs are anyhow confusing since in 1C there is a band in the ctrl while there is no GFP-STIM1 present

This statement is not correct. The control for the colP shown in Figure 1C does not have GFP-STIM1, but the empty GFP vector. We will include the lower weight gel band.

3. In addition, Fig2C: How come the IPed band for p150glued runs higher than p150glued in the TL? Also the position of protein marker standards should be shown.

This statement is not correct. Fig. 2C is a graph illustrating the quantification of LE number.

- The authors show (most of the time) images with only one phenotypic cell. Seeding the cells a bit more dense will allow them to show multiple cells at ones which will strengthen their conclusions.

Cells were already subconfluent. As already discussed above, zooming in would no longer allow to appreciate heterogeneities in the distribution of endosomes with the cell. On the contrary zooming out will make almost impossible to see endosomes. Culturing cells at higher confluence would make difficult to distinguish cell boundaries (as the reviewer also stated above).

- The authors exclude clustered endosomes for their analyses. Is it fair to do this? In control situation the majority of LEs are clustered, are these all excluded? For example in Fig2B they analysed a total of 150 endosomes from 30 cells. That means 5 endosomes per cell? I guess I misunderstood this, but then correct. Also STIM1 depletion leads to more dispersed endosomes which the authors conclude are smaller in size compared to endosomes in the control. Endosomes are known to be smaller in the periphery/ or enlarge a bit due to fusion events if they reach the PN area (as the authors also point out). If they exclude all the clustered endo's (most likely bigger ones), they may automatically affect endosomal size, which would then not be an informative result.

Concerning the number of EE considered, **we will extend, as mentioned before, the automated method, used to quantify early endosome parameters in live imaging experiments as in Fig 6B**, also to fixed staining of both late and early endosomes in rescued experiments in Figures 2,3,4,5 and 7.

Concerning the exclusion of the perinuclear area to quantify late endosome, it is true that we exclude the majority of endosomes in the control situation, but **we exclude the same area in the si/shSTIM1 or control treated conditions**. Doing that, we see an increase in number of LEs in the experimental situations, with a decreased size, which excludes fusion events between those smaller vesicles.

- It is misleading to use SEM for (WB) quantifications., this results in super small and unrealistic error bars. The authors should use SD instead and preferably plot single values.

We will change SEM for the SD in WB analysis.

- Remaining questions: what determines if Dynein or KIFC1 is in charge of EE transport? Are Hook3 and Hook 1 on similar or distinct EE vesicles? This part of the paper is very limited while this part actually contains their main and most interesting message.

As experimentally shown by our data and summarized in the model shown in Figure 8B, the transport of early endosomes require the engagement of both Dynein:p150glued:HOOK1 and KIF1C:p150glued:HOOK3 motor systems transported along MTs. If only one of the retrograde transport systems is engaged early endosomes are transported too fast towards the perinuclear area, where they accumulate.

Other comments:

- To deplete cells for STIM1 they use a pool of siRNAs, but they see a similar phenotype using shRNA (so they do use multiple).

We agree with the reviewer. **We must however emphasize that this statement contradicts her/his sixth statement above**, where s/he wrote “However, a more substantial problem (and that is throughout this manuscript) is that STIM1 is only silenced with one shRNA or a pool of siRNA and off target effects can then not be excluded. This is not the standard in cell biology tmo.”

Referee #3:

In this manuscript, Villari et al report an interaction between the endoplasmic reticulum (ER) transmembrane protein stromal interaction molecule 1 (STIM1) and p150Glued, a subunit of the dynactin complex. They show that STIM1 promotes the localization of p150Glued at the microtubule (MT) plus ends through the formation of a ternary complex with the MT end-binding protein 1 (EB1). The authors then investigate the role of STIM1 in the distribution of early (EEs) and late endosomes/lysosomes (LEs). Silencing of STIM1 leads to opposite effects, LEs being redistributed toward the cell periphery whereas peripheral EEs are clustered around the nucleus. In a second part of the manuscript, the authors further explore the mechanisms responsible for the peripheral localization of EEs. They provide evidence that their localization relies on the antagonistic action between dynein and the MT minus-end motor KIFC1, which are attached to EEs via HOOK1 and HOOK3 adaptor complexes, respectively.

STIM1 is an ER Ca²⁺ sensor that plays a key role in store-operated calcium entry (SOCE) by interacting with the plasma membrane (PM) Ca²⁺ channel Orai1 following depletion of ER Ca²⁺ stores. STIM1 interaction with EB1 is thought to regulate its translocation to PM-ER contact sites. That STIM1 is also involved in the distribution and transport of endosomal compartments is a novel finding of potential interest. Nevertheless, the manuscript suffers from major weaknesses and in its present form, does not meet the necessary standards for a publication in EMBO J.

Major comments:

- 1) Figure 1: The authors claim that STIM1/p150Glued/EB1 forms a ternary complex. This is not directly proven, Figs 1A-C simply showing protein interactions two by two. In addition, experiments were performed following overexpression of tagged STIM1 and p150Glued. Did the authors try to co-IP endogenous proteins?

The reviewer is absolutely right, we didn't try to co-IP the three endogenous proteins together and we will definitely do it.

The results presented in Fig 1C and 1G seem contradictory. In Fig.1C, the NN STIM1 mutant (unable to bind EB1) can be co-IP with p150Glued but does not co-localize with it by IF (Fig. 1G).

The reviewer is right in the analysis of data. NN STIM1 binds p150Glued, but not MT plus end associated protein EB1. Yet, rather than being contradictory, these data suggest a model in which, even if NN STIM1 can bind p150Glued, the inability to simultaneously bind EB1 impedes NN STIM1 to favor the enrichment of p150Glued at EB1 containing MT plus ends and consequently the co-localization of NN STIM1 and p150Glued at MT plus ends. These data are indeed compatible only with a ternary complex formation model, drawn in Figure 1D.

- 2) Figure 2: The authors conclude for a dual role of STIM1 in SOCE and LE distribution based on the rescued effect of the AA STIM1 mutant (unable to bind Orai1) (Figs 2E, F). This is not enough. The authors should investigate the effect of the depletion of ER calcium stores on the distribution of LEs. These experiments are also important to rule out the possibility that the effects of STIM1 depletion on LE distribution result from changes in ER morphology.

We totally agree with the reviewer on this point and we did already test that. Indeed, while we did not show it, we already tested this hypothesis and found that treating cells with Thapsigargin (a well-known inhibitor of the SERCA ER ATPase, which induces ER Ca²⁺

store depletion) did not show any effect on late endosome positioning. **We will include the data, which we already obtained, in the manuscript.**

3) Figures 5 and 6: The localization of KIFC1 should be provided. Based on the effects of silencing both KIFC1 and STIM1 or inhibiting KIFC1 and dynein, the authors suggest that KIFC1 and STIM1/dynein "antagonistically" cooperate. KIFC1 and dynein/dynactin complex being both MT-end motors, how can it work? No mechanism(s) is (are) proposed.

We agree that KIFC1 localization is missing and we will provide it, but we already did propose a mechanism in the submitted manuscript. Indeed, we think that the HOOK adaptors are central for the engagement of the two motors. As drawn in the model in Figure 8B, early endosomes require the engagement of both Dynein:p150glued:HOOK1 or KIF1C:p150glued:HOOK3 motor systems to be retrogradely transported along MTs.

4) Figure 7: IP experiments shown in Fig. 7A-C are incomplete. Does HOOK3 antibody coimmunoprecipitate KIFC1? Why KIFC1 was not probed in Fig. 1A? Fig. 7E should be quantified.

Yes, as already shown in Figure 7C, HOOK3 antibody does coimmunoprecipitate with KIFC1 together with HOOK3. We will also add KIFC1 analysis into panel A of Fig. 1. We will also quantify 7E.

5) Figure S3C: The hypothesis that STIM1/dynein compete for binding to p150Glued is interesting but should be supported by stronger results. Does KIFC1 silencing affect binding of p150Glued to HOOK1?

We agree with the reviewer. We have never tried that and we will.

6) It is written in almost all Fig. legends that quantification of endosomes in fixed cells were performed "for a total of 150 endosomes (30 cells)". Does it mean that only 5 endosomes have been analyzed per cell?

We will extend the automated method, used to quantify early endosome parameters in live imaging experiments as in Fig 6B, also to fixed staining of both late and early endosomes in rescued experiments in Figures 2,3,4,5 and 7.

Other comments:

7) The contribution of centrosomal MTs to EE localization should be at least discussed. A + end kinesin motor, KIF16B, was shown to modulate the distribution of EEs (Hoepfner et al, Cell 2005).

This point is definitely relevant for a more complete understanding of the mechanism. So, we will silence KIF16B singularly and in combination with both dynein and KIFC1 inhibitors to check early endosome positioning.

8) Figure 1C: Why does the GFP-STIM CC mutant (deletion of the 3 coiled coil domains) show exactly the same electrophoretic motility than WT? In Fig. S1, delta CC2 and CC3 mutants migrate faster than WT.

We will check the blot shown in Fig 1C.

9) The authors claim that they verify "the ability of ER protein STIM1 to bind, similarly to p150Glued, EB1 (Grigoriev et al. 2008). The STIM1-p150Glued interaction has not been documented in that manuscript.

The reviewer is right, we confused the references. Apologies. We meant Honnappa et al. Cell 2009 to say that STIM1, p150Glued and EB1 are all known microtubule +TIP proteins.

Thank you again for contacting me with a preliminary point-by-point response outlining the scope of a potential revision of your manuscript. I apologise for replying to you with such a delay due to the high submission rate to our office, since new submissions are treated with priority in these cases.

I appreciate from your revision plan that you are willing to engage in an extensive revision to address the main issues raised by the reviewers, and already have data to clarify some of the points (as a minor comment, I think that reviewer #1 in their point 3 is suggesting that potential formation of hybrid organelles with mixed early/late endosome identity would have to be tested). However, since the indicated issues are rather far-reaching, affect core conclusions of the manuscript, and appear to be out of scope of our usual 3 month revision period, I am afraid that I cannot explicitly invite a revised manuscript.

Nevertheless, if you find that you can address all main referee concerns and provide substantial additional support to the proposed mechanism, I would be happy to reconsider the manuscript, while treating it as a new submission. In this case, it means that I would send it back to the same reviewers, if possible, but would allow them to make new comments on the data, which might then have to be further addressed if the reviewers are more positive in this round of assessment.

Referee #1:*Major points:*

1. *The present data do not provide a plausible explanation for the differential localization of early and late endosomes, especially since both dynein and KIF1C are minus-end-directed microtubule motors. The authors need to study whether plus-end-directed motors such as KIF2 and KIF5 also associate preferentially with one of the two endosome populations. If so, then endosome positioning could equally well be explained by alternative mechanisms.*

As proposed by the Reviewer, we investigated **KIF5B**, which was reported as a microtubule (MT) plus-end directed kinesin modulating early endosome (EE) motility and traffic (Braun et al., 2016, *Curr Biol* 26: R1292-R1294; Nath et al., 2007, *Mol Biol Cell* 18: 1839-1849; Loubéry et al., 2008, *Traffic* 9: 492-509), as a possible EE anterograde motor to counterbalance the dynein/KIF1C-driven minus-end transport system that we identified. In agreement with Reviewer hypothesis, as described on **page 12** and shown in **Fig. S2E**, we found **KIF5B silencing to impair the EE peripheral distribution induced by simultaneous CilioD and AZ82 treatment**, suggesting its engagement when both EE retrograde transport motors, Dynein and KIF1C, are inhibited.

2. *The functional implications of the reported effects on endosome distributions remain unclear. The authors need to test effects on biological activities that have been reported to depend on endosome positioning, such as cargo degradation, recycling and mTORC1 signaling.*

As suggested by the Reviewer, we identified and studied possible **functional implications of STIM1-dependent endosome distribution**. It has been previously shown that subcellular positioning controls late endosome (**LE**) **luminal pH** (Johnson et al., 2016, *J Cell Biol* 212: 677-692) and the **activation of mTORC1** on LE cytosolic surface (Korolchuk et al., 2011, *Nat Cell Biol* 13: 453-460). Furthermore, activated mTORC1 inhibits **autophagy** (Korolchuk et al., 2011, *Nat Cell Biol* 13: 453-460). Hence, we decide to evaluate the functional impact of STIM1 silencing on LE pH, mTORC1 signaling and autophagy. As described in the new Results section "STIM1 controls LE pH, mTORC1 signaling and autophagy" at **pages 14-15** and shown in **Fig. 8**, we found that LE peripheral scattering caused by **STIM1 silencing** correlated with **increased LE pH**, activation of **mTORC1 signaling** downstream effectors (phosphorylation of **S6K1/p70**), and **decreased autophagy**, thus supporting a functional role of STIM1 in the control of LE positioning dependent pH, response to nutrients and autophagy. The potential implications are also discussed at **page 17**.

3. The authors should consider the possibility that interference with motor functions might affect endosome maturation. Using EEA1 as an early endosome marker and LAMP1 as a late endosome marker might be confounded if motor depletion affects endosome maturation.

In **Fig. 3** of the first version of our manuscript, we showed that the effect of STIM1 silencing on EE positioning was seen using **both anti-EEA-1** and **anti-Rab5** antibodies, suggesting that the two markers are comparable to observe the phenotype. At present, we could not perform a similar parallel staining for LAMP-1 and Rab7, as we were unable to identify any reliable Rab7 antibody among the many tested so far.

To address the Reviewer point on endosome maturation, as described on **page 10** and **Fig. S2A**, we quantified **Pearson correlation between EEA-1⁺ and LAMP-1⁺ endosomes** in control and STIM1 silenced ECs as a measure of the **fraction of EEs en route to transformation in LEs**. The studied **silencing of STIM1 does not affect** the Pearson correlation between EEA-1⁺ and LAMP-1⁺ endosomes, supporting the concept that siSTIM1-induced interference on motor function does not detectably impact on EE-to-LE maturation.

Minor point:

Bar diagrams should be replaced by scatter plots, which give a visualization of data distribution.

All bar plots were replaced with scatter plots.

Referee #2:

1. The experiments depend on microscopy analyses of endosomes in cells silenced for the different proteins, sometimes overexpression of fragment of these proteins and cIP experiments followed by WB for the analyses of interactions to proteins or their domains. They are not always very convincing. For example, Fig 1 shows the interaction between STIM1 and p150 by cIP/WB and by confocal. Fig 1b shows a very poor cIP between overexpressed STIM1 and p150 that is 1 band. Fig 1c shows the same interaction with STIM1 fragments but now reveals 2(!) p150 bands.

In Fig. 1C of the first version of our manuscript, we showed co-immunoprecipitation of **overexpressed** mCherry-p150Glued together with **overexpressed** GFP-tagged STIM1 mutants. The double p150Glued bands were due the fact that we employed, to detect the immunoprecipitated with GFP-STIM1, an **anti-p150Glued antibody**, that recognized both endogenous p150Glued and overexpressed mCherry-p150Glued.

We agree with the Reviewer that previous Fig. 1C was misleading. Therefore, as described on **page 7** and shown in the **new Fig. 1E** of the revised manuscript, we repeated the coimmunoprecipitations on lysate of cell overexpressing mCherry-p150Glued and GFP-tagged STIM1 mutant constructs in which we employed an **anti-mCherry antibody**, which immunopurified overexpressed mCherry-p150Glued only.

Furthermore, as described on **page 6** and shown in **new Fig. 1B**, **now** we show how in endothelial cells **endogenous p150Glued** co-immunoprecipitates with **endogenous STIM1**.

2. Also the microscopy in Fig 1e shows that p150 decorated microtubules and two selected dots with some colabelling (not that the blob is different). However, magnification of the figures shows that the co-labelling is in fact very poor.

Images of Fig. 1E of the first version of our manuscript (now **Fig. 1G**), in agreement with what previously described by the laboratory of Anna Akhmanova (Grigoriev et al., 2008, *Curr Biol* 18: 177-82), showed that:

- a) **STIM1 WT** and **ΔCC/WT**, which **bind EB1**, enriched in punctate structures **at MT plus ends**, where overexpressed **p150Glued** is also **enriched**;
- b) **STIM1 NN**, which lacks the SxIP motif and **does not bind to EB1**, results in an **evident** overall **diminishment** of the **p150Glued** positive punctate structures with a **decreased Pearson** correlation between **p150Glued** and **STIM1**.

To make the new **Fig. 1G** easier to be observed, now we provide **close-up** panels of **wider cellular**

areas that allow to better appreciate the **multiple colocalizations** between p150Glued and STIM1 within those punctate structures that we also highlighted by means of **arrows**.

As in our previous version of the manuscript, we **quantified**:

- a) **p150Glued positive punctate structures** in **number** (**new Fig. 1H**);
- b) **colocalization** between **p150Glued** and **GFP-STIM1** WT or its mutants quantified as **Pearson** correlation, shown in **new Fig. 1I**.

Altogether, these **microscopy** data (described on **page 8**) further highlight the key **functional implications of the triple STIM1/EB1/p150Glued complex formation model**, already observed **biochemically** in coimmunoprecipitations (shown in the **new Fig. 1E**) and further supported by our **novel in vitro interaction study** (see **reply to Reviewer point 3 below**). Indeed, the STIM1-elicited formation of p150Glued positive structures strictly relies on the simultaneous binding of STIM1 to EB1, in addition to STIM1 direct interaction with p150Glued.

3. It would also be helpful if the authors test the interaction with purified domains.

Following Reviewer suggestions, we performed **in vitro interaction assays**, like those set up by the laboratory of Thomas Surrey (Duellberg et al., 2014, *Nat Cell Biol* 16:804-811), by employing the following purified proteins (depicted in **new Fig. 1C**):

- a) whole **cytoplasmic portion of STIM1 (STIM1 cyto)** with intact EB1-binding SxIP motif;
- b) **N-terminal portion of p150Glued** containing the **Cap-Gly domain** (known to mediate the binding to the EB1 C-terminal EEY motif) and first coiled coil (CC) **domain (CC1a) (Cap-Gly-CC1a-p150Glued)**. In addition to the N-terminal EB1-binding Cap-Gly domain of p150Glued, we also included in our purified p150Glued construct the first CC domain, CC1a, because we posited it would have been the only one, among the three CC regions of p150Glued (Tripathy et al., 2014, *Nat Cell Biol* 16:1192-1201), available for the binding to the CC1-3 domains of STIM1, being the second (CC1b) and third (CC2) domains of p150Glued known to interact with dynein IC and the Arp1 filament of dynactin, respectively (McKenney, 2018, *Dyneins: The Biology of Dynein Motors*; Reck-Peterson et al., 2018, *Nat Rev Mol Cell Biol* 19: 382-398). Moreover, since p150Glued CC1a domain is known to exist in an inhibited form folded with the second CC1b domain, excluding CC1b domain of p150Glued would have allowed us to avoid any functional inactivation of p150Glued CC1a domain (Want et al., 2014, *Proc Natl Acad Sci U S A* 111: 11347-11352; Saito et al., 2020, *Mol Biol Cell*:

mbcE20010031).

c) **C-terminal portion of EB1** containing the STIM1-binding **EBH domain** and the p150Glued-interacting **EEY motif**. To assess the importance of the involvement of EB1 EEY motif binding to p150Glued Cap-Gly domain (Akhmanova & Steinmetz, 2015, *Nat Rev Mol Cell Biol*, 16: 711-726) in the formation of the ternary complex with STIM1, we generated and tested two different constructs of the C-terminal portion of EB1. A first wild type (**EB1 WT C-term**) EEY motif-containing construct and a second construct in which the last C-terminal tyrosine residue of the EEY motif was deleted (**EB1 Δ Y C-term**).

As described on **pages 6-8** and shown in **Fig. 1C, 1D** and **1F**, we: i) confirmed *in vitro* the binding between purified Cap-Gly-CC1a-p150Glued and STIM1 cyto; ii) found that the *in vitro* interaction between Cap-Gly-CC1a-p150Glued and STIM1 cyto was clearly stabilized by the addition of purified EB1 WT C-term, but not EB1 Δ Y C-term.

To sum up, **we confirmed *in vitro* the interaction of affinity purified STIM1, p150Glued and EB1 constructs** to give rise to a triple complex. We also confirmed the need of a cooperation between the EBH domain and the EEY motif of EB1 to stabilize the interaction between STIM1 and p150Glued.

4. *In figure 2 the authors quantify the late endosome size, which is remarkable given the quality of the pictures.*

Images were acquired with a Leica TCS SP8 AOBS confocal laser-scanning microscopes (Leica Microsystems, Wetzlar, Germany). A PL APO 100x/1.4 NA immersion objective was employed. 1024x1024 pixel images were acquired at **pixel size = 87,30-92,26 nm**.

We agree with the Reviewer (see also her/his points 7 and 12) that our original choice of excluding the perinuclear clustered endosomes, particularly for LE control situation, could be misleading and underestimated our previous counts. To fix this issue for all quantified images, we sought to generate a **maximum intensity projection from a z-stack of sections spanning the whole perinuclear area and acquired with a step $\leq 0.37 \mu\text{m}$. This approach allowed us to increase the resolution in the perinuclear area and distinguish singular perinuclear objects. Moreover, our new method includes two levels of threshold (described on pages 24-25) on those objects, being selected in terms of both singularity and shape**. We included an example of the resulting masks for both LEs and EEs in **Fig. S3E** of the revised manuscript. Doing so, as also discussed in our reply to Reviewer's point 5, we significantly increased the number of endosomes per cell counted.

To favor the same quality of observation of endosome with different subcellular distribution in the whole cell and appreciate changes in their distance to the nucleus together with a closer view on modifications in size and number of the same vesicles, we added **magnified close-ups of both perinuclear and peripheral areas of the cell to each panel of Fig. 2-7, S1 and S2.**

5. How the authors arrive at the extremely small SEM in their quantification, is surprising, especially when measuring such small endosomes.

In the first version of this manuscript endosomes we automatically quantified only in live CilioD and AZ82 treatment experiments. We extended our **novel automated quantification method** (see *Materials & Methods* section, pages 23-25) to detect **endosome number, size and distance to nucleus to all the conditions** evaluated in the revised version of our manuscript. Our automated quantification method proved to be sensitive to variations in size and distance of endosomes, otherwise hard to detect, providing large amount of data and reliable small SEM. We acquired **50-200 LEs or EEs per cell in at least 15 cells** (getting to **≈750-2000 endosomes in total**) **per experiment**. Raw data, as *per lab* and *EMBOJ.* policy, were collected and organized in a dedicated *Excel* file, which is provided as supplementary material.

6. However, a more substantial problem (and that is throughout this manuscript) is that STIM1 is only silenced with one shRNA or a pool of siRNA and off target effects can then not be excluded. This is not the standard in cell biology tmo.

As highlighted by the reviewer her/himself at point 15, **we used a pool of siRNAs and one shRNA** throughout the manuscript and we observed similar phenotypes.

7. A cell is a 3D system. However, the authors only show data from 2D images. Endosomes are typically clustered in the perinuclear area, however the image plane will largely affect the number of imaged endosomes as well as their localization (peripheral endosomes are mainly found closer to the glass surface). The authors should consider analysing the whole cell (3D) or at least indicate that the image plane is constant/similar between all conditions shown. No cell boundaries are indicated in any of the pictures (or I might fail to see it on the print/screen).

We agreed with the Reviewer (see also her/his points 4 and 12) that our original choice of excluding the

perinuclear clustered endosomes, particularly for LE control situation, could be misleading and underestimated our previous counts. To fix this issue for all quantified images, we sought to generate a **maximum intensity projection from a z-stack of sections spanning the whole perinuclear area and acquired with a step $\leq 0.37 \mu\text{m}$. This approach allowed us to increase the resolution in the perinuclear area and distinguish singular perinuclear objects. Moreover, our new method includes two levels of threshold (described on pages 24-25) on those objects, being selected in terms of both singularity and shape.** We included an example of the resulting masks for both LEs and EEs in **Fig. S3E** of the revised manuscript. Doing so, as also discussed in our reply to Reviewer's point 5, we significantly increased the number of endosomes per cell counted.

Additionally, as further explained in the *Materials and Methods* section (page 25-26), in the live imaging quantification of Rab5⁺ EEs:

- the **confocal image thickness along z axis** was kept **constant** at **0,9 μm** among all conditions. This value results particularly relevant using endothelial cells, since they are flat and wide cells whose thickness is about 1 μm in the area away from the nucleus;
- we considered **a single endosome an object appearing consecutively for 60 seconds in the silencing experiments and for 30 second for the treatment ones.** Through such method, when an endosome disappears and another one appears in its same position after 2 frames, the second one is considered a "new endosome". Moreover, we selected those **EEs spatially limited within 5 pixels ($\approx 500 \text{ nm}$)**, to better represent the Brownian motion of those vesicles. That stringent thresholding method further implemented our quantification.

Moreover, to make changes in EE or LE positioning in relation to the whole cell clearer, as suggested by the Reviewer, **we added cell boundaries in the immunofluorescence pictures.**

8. The IP controls are rather unusual. They incubate lysates with empty beads as control for endogenous immunoprecipitation. If the authors would like to use such controls they should at least coat the beads with isotype controls/serum. Better would be to just compare the control with the depleted situation (which they also did occasionally).

We repeated the immunoprecipitation experiments, lacking with **proper controls**, incubating beads with **IgG of the same species of the primary antibody** used for the immunoprecipitation.

9. *Their overexpression WBs lack proper washing controls. They incubate lysates (for example GFP/RFP transfected) with empty beads. They should include the Ab used for immunoprecipitation also in this sample to show that the co-immunoprecipitated proteins does NOT bind under EV conditions. Their WBs are anyhow confusing since in 1C there is a band in the ctrl while there is no GFP-STIM1 present*

The control for the coimmunoprecipitation shown in previous Fig. 1C (now **Fig. 1E**) was done incubating lysates of cells expressing mCherry-p150Glued together with a GFP-only vector with the anti-GFP used in the coimmunoprecipitation experiment. For sake of clarity, **we added the lower molecular weight gel part** (showing 25 kDa GFP).

10. *In addition, Fig2C: How come the IPed band for p150glued runs higher than p150glued in the TL? Also the position of protein marker standards should be shown.*

We repeated the coimmunoprecipitation and Western blot analyses of previous Fig. 1C (now **Fig. 1E**), see also our reply to Reviewer point 1.

11. *The authors show (most of the time) images with only one phenotypic cell. Seeding the cells a bit more dense will allow them to show multiple cells at ones which will strengthen their conclusions.*

As discussed in our reply to Reviewer's point 4 and 5, we chose to show one cell per image (representative of the phenotype of a large number of endosomes and cells considered) **to allow to simultaneously appreciate heterogeneities in EE or LE distribution within the cell, but also changes in their size and number**, which would have been not possible to appreciate in lower power images with multiple cells. Moreover, culturing cells at a higher confluence would have made difficult to distinguish cell boundaries (as also the reviewer stated above). As mentioned in our reply to Reviewer's point 4 and 7, **we added, to all immunofluorescence panels, close-ups of both perinuclear and peripheral areas of the cell together with cell boundaries.**

12. *The authors exclude clustered endosomes for their analyses. Is it fair to do this? In control situation the majority of LEs are clustered, are these all excluded? For example in Fig2B they analysed a total of 150 endosomes from 30 cells. That means 5 endosomes per cell? I guess I misunderstood this, but then correct. Also STIM1 depletion leads to more dispersed endosomes which the authors conclude are smaller in size compared to endosomes in the control. Endosomes are known to be smaller in the periphery/or enlarge a bit*

due to fusion events if they reach the PN area (as the authors also point out). If they exclude all the clustered endo's (most likely bigger ones), they may automatically affect endosomal size, which would then not be an informative result.

We agree with the Reviewer (see also her/his points 4 and 7) that our original choice of excluding the perinuclear clustered endosomes, particularly for LE control situation, could be misleading and underestimated our previous counts. To fix this issue for all quantified images, we sought to generate a **maximum intensity projection from a z-stack of sections spanning the whole perinuclear area and acquired with a step $\leq 0.37 \mu\text{m}$. This approach allowed us to increase the resolution in the perinuclear area and distinguish singular perinuclear objects. Moreover, our new method includes two levels of threshold (described on pages 24-25) on those objects, being selected in terms of both singularity and shape.** We included an example of the resulting masks for both LEs and EEs in **Fig. S3E** of the revised manuscript.

Furthermore, as also discussed in our reply to Reviewer's point 5, we significantly increased the number of endosomes per cell counted. Indeed, in the first version of this manuscript endosomes we automatically quantified only in live CilioD and AZ82 treatment experiments. In this revised version, we extended our **novel automated quantification method** (see *Materials & Methods* section, **pages 24-25**) to detect **endosome number, size and distance to nucleus to all the conditions** evaluated in the revised version of our manuscript. Our automated quantification method proved to be sensitive to variations in size and distance of endosomes, otherwise hard to detect, providing large amount of data and reliable small SEM. We acquired **50-200 LEs or EEs per cell in at least 15 cells** (getting to **$\approx 750-2000$ endosomes in total**) **per experiment**.

13. It is misleading to use SEM for (WB) quantifications., this results in super small and unrealistic error bars. The authors should use SD instead and preferably plot single values.

We changed to SD all the SEM in Western blot and Pearson correlations analyses and used **scatter plots**.

14. Remaining questions: what determines if Dynein or KIFC1 is in charge of EE transport? Are Hook3 and Hook 1 on similar or distinct EE vesicles? This part of the paper is very limited while this part actually contains their main and most interesting message.

As experimentally shown by our data (**Fig. 7** and **Fig. S3**) and summarized in the schematic model shown in **Fig. 9B**, the transport of most EEs requires the **simultaneous engagement** of both **Dynein:p150glued:HOOK1** and **KIFC1:p150glued:HOOK3** motor systems transported along MTs towards . If only one of those retrograde

transport systems is engaged, EEs are transported faster towards the perinuclear area, where they accumulate.

Other comments:

15. To deplete cells for STIM1 they use a pool of siRNAs, but they see a similar phenotype using shRNA (so they do use multiple).

As already stated in our reply to point 6, we agree with the Reviewer: we used a pool of siRNAs and one shRNA throughout the manuscript, so **we did use multiple silencing strategies**, and we observed similar phenotypes.

Referee #3:

Major comments:

1. *Figure 1: The authors claim that STIM1/p150Glued/EB1 forms a ternary complex. This is not directly proven, Figs 1A-C simply showing protein interactions two by two. In addition, experiments were performed following overexpression of tagged STIM1 and p150Glued. Did the authors try to co-IP endogenous proteins?*

We agree with the Reviewer. As described on page 6 and shown in new Fig. 1B, we repeated the experiments by immunoprecipitating the three endogenous protein together.

Moreover, we performed in vitro interaction assays, like those set up by the laboratory of Thomas Surrey (Duellberg et al., 2014, *Nat Cell Biol* 16:804-811), by employing the following purified proteins (depicted in new Fig. 1C):

a) whole cytoplasmic portion of STIM1 (STIM1 cyto) with intact EB1-binding SxIP motif;

b) N-terminal portion of p150Glued containing the Cap-Gly domain (known to mediate the binding to the EB1 C-terminal EEY motif) and first coiled coil (CC) domain (CC1a) (Cap-Gly-CC1a-p150Glued). In addition to the N-terminal EB1-binding Cap-Gly domain of p150Glued, we also included in our purified p150Glued construct the first CC domain, CC1a, because we posited it would have been the only one, among the three CC regions of p150Glued (Tripathy et al., 2014, *Nat Cell Biol* 16:1192-1201), available for the binding to the CC1-3 domains of STIM1, being the second (CC1b) and third (CC2) domains of p150Glued known to interact with dynein IC and the Arp1 filament of dynactin, respectively (McKenney, 2018, *Dyneins: The Biology of Dynein Motors*; Reck-Peterson et al., 2018, *Nat Rev Mol Cell Biol* 19: 382-398). Moreover, since p150Glued CC1a domain is known to exist in an inhibited form folded with the second CC1b domain, excluding CC1b domain of p150Glued would have allowed us to avoid any functional inactivation of p150Glued CC1a domain (Want et al., 2014, *Proc Natl Acad Sci U S A* 111: 11347-11352; Saito et al., 2020, *Mol Biol Cell*: mbcE20010031).

c) C-terminal portion of EB1 containing the STIM1-binding EBH domain and the p150Glued-interacting EEY motif. To assess the importance of the involvement of EB1 EEY motif binding to p150Glued Cap-Gly domain (Akhmanova & Steinmetz, 2015, *Nat Rev Mol Cell Biol*, 16: 711-726) in the formation of the ternary complex with STIM1, we generated and tested two different constructs of the C-terminal portion of EB1. A first wild type (EB1 WT C-term) EEY motif-containing construct and a second construct in which the last C-terminal

tyrosine residue of the EEY motif was deleted (**EB1 Δ Y C-term**).

As described on pages 6-8 and shown in Fig. 1C, 1D and 1F, we: i) confirmed *in vitro* the binding between purified Cap-Gly-CC1a-p150Glued and STIM1 cyto; ii) found that the *in vitro* interaction between Cap-Gly-CC1a-p150Glued and STIM1 cyto was clearly stabilized by the addition of purified EB1 WT C-term, but not EB1 Δ Y C-term.

To sum up, **we confirmed *in vitro* the interaction of affinity purified STIM1, p150Glued and EB1 constructs** to give rise to a triple complex. We also confirmed the need of a cooperation between the EBH domain and the EEY motif of EB1 to stabilize the interaction between STIM1 and p150Glued.

2. The results presented in Fig 1C and 1G seem contradictory. In Fig.1C, the NN STIM1 mutant (unable to bind EB1) can be co-IP with p150Glued but does not co-localize with it by IF (Fig. 1G).

We agree with the Reviewer. STIM1 NN still binds p150Glued, but not MT plus end associated protein EB1. Yet, we propose that, rather than being contradictory, these data suggest a model in which, even if that STIM1 mutant can bind p150Glued, **the inability to simultaneously bind EB1 impedes STIM1 NN to favor the enrichment of p150Glued** at EB1 containing MT plus ends and consequently its co-localization with STIM1. These data indeed support the **ternary complex formation model**, drawn in Fig. 1C and verified in our **novel *in vitro* protein-protein interaction studies** described in our reply to point 1 of the Reviewer.

3. Figure 2: The authors conclude for a dual role of STIM1 in SOCE and LE distribution based on the rescued effect of the AA STIM1 mutant (unable to bind Orai1) (Figs 2E, F). This is not enough. The authors should investigate the effect of the depletion of ER calcium stores on the distribution of LEs. These experiments are also important to rule out the possibility that the effects of STIM1 depletion on LE distribution result from changes in ER morphology.

We totally agree that the effect of the depletion of ER calcium stores on LE distribution was an aspect to be further investigated. Therefore, we treated cells with **Thapsigargin** [a well-known inhibitor of the Sarco-Endoplasmic Reticulum Calcium ATPase (SERCA)], which induces ER Ca²⁺ store depletion and **STIM1 detachment from EB1 at microtubule (MT) plus end**, thus causing its association at the plasma membrane with the calcium channel subunit ORAI1, allowing the assembly of the store-operated calcium influx complex that replenishes the ER Ca²⁺ stores (Vaca, 2010, *Cell Calcium*, 47:199-209). Of note, as described on page 9 and shown in Fig. S1E and S1F, when we incubated cells with Thapsigargin (which detaches STIM1 from MTs)

and measured LE distance to the nucleus with our automated method (\approx 450 endosomes counted per condition), we found that **LE scatter towards the cell periphery**, similarly to what observed in STIM1 silenced cells. Together with the STIM1 AA mutant rescue experiments (already shown in Fig. 2), those data with Thapsigargin further support a dual role for STIM1 as both **ER Ca²⁺ store sensor** and **dynein dependent LE positioning modulator**.

4. Figures 5 and 6: The localization of KIFC1 should be provided. Based on the effects of silencing both KIFC1 and STIM1 or inhibiting KIFC1 and dynein, the authors suggest that KIFC1 and STIM1/dynein "antagonistically" cooperate. KIFC1 and dynein/dynactin complex being both MT-end motors, how can it work? No mechanism(s) is (are) proposed.

Although several attempt to visualize KIFC1 (both endogenous proteins, with antibodies, and overexpressed constructs, by transfection of exogenous GFP-tagged cDNAs), **we could not observe by fluorescence confocal microscopy a neat and clear subcellular localization of KIFC1** in either fixed and stained or living cells.

However, we provided **new evidences** (described on **page 13** and shown in **Fig. 7H**, see also below point 6) on the already presented model (**Figs. 7 and 9**) of the **concerted antagonistic activity** of STIM1/Dynein and KIFC1 motors in the regulation of EE retrograde transport. Indeed, in addition to the central role of HOOK adaptors in the simultaneous engagement of the specific motor (**Dynein:p150glued:HOOK1** or **KIFC1:p150glued:HOOK3**), we performed and show in the revised version of our manuscript **a new set of experiments** that support a **competition between the two motor systems to interact with the MT plus end protein EB1 (Fig. S3D)**.

5. Figure 7: IP experiments shown in Fig. 7A-C are incomplete. Does HOOK3 antibody coimmunoprecipitate KIFC1? Why KIFC1 was not probed in Fig. 1A? Fig. 7E should be quantified.

As suggested by the Reviewer, to **better characterize the interaction specificity of HOOK adaptors** to the respective motor system, in **Fig. 7A** the **HOOK1 immunoprecipitate** was analyzed in **Western blot** also with an **anti-KIFC1** antibody. On the contrary, in Fig. 7C of the first version of the manuscript (corresponding to **Fig. 7B** in the new revised version) the **KIFC1 immunoprecipitate** had **already** been analyzed in **Western blot** with an **anti-HOOK3** antibody.

As requested, we also **quantified both EE (Fig. 7E-G) and LE (Fig. S3B, C)** distance to the nucleus in **HOOK1**

or HOOK3 silenced cells.

6. Figure S3C: The hypothesis that STIM1/dynein compete for binding to p150Glued is interesting but should be supported by stronger results. Does KIFC1 silencing affect binding of p150Glued to HOOK1?

We are grateful to the Reviewer for this stimulating question. As described on page 13 and shown in Fig. 7H, we tested whether KIFC1 inhibition may affect binding of p150Glued to HOOK1, confirming the hypothesis of a competition between HOOK adaptors for p150Glued binding in the engagement of the dual EE motor systems. We immunoprecipitated p150Glued from endothelial cells treated or not with the KIFC1 specific inhibitor AZ82 and verified its association with HOOK1 adaptor by Western blot. Interestingly, we found that AZ82 inhibition of KIFC1 activity strengthens the binding of HOOK1 to p150Glued.

Moreover, as described on page 13 and shown in Fig. S3D, supporting a model of competition between the SxIP motif of STIM1 (associated to dynein) and KIFC1 for the binding to the EBH domain of EB1 at MT plus ends, we also found that the lack of STIM1 increased the binding of KIFC1 to EB1 in ECs.

Altogether, these data sustain a two-layered competition model between STIM1/dynein and KIFC1 to interact with the (i) p150Glued subunit of dynactin and (ii) EB1 to cooperatively regulate EE retrograde transport.

7. It is written in almost all Fig. legends that quantification of endosomes in fixed cells were performed "for a total of 150 endosomes (30 cells)". Does it mean that only 5 endosomes have been analyzed per cell?

We extended the automated quantification method to detect endosome number, size and distance to nucleus (previously limited to live CilioD and AZ82 treatment experiments only) to all the conditions evaluated in the manuscript. Doing so, we acquired 50-200 LEs or EEs x at least 15 cells (getting to ~750-2000 endosomes in total) per experiment.

Other comments:

8. *The contribution of centrosomal MTs to EE localization should be at least discussed. A + end kinesin motor, KIF16B, was shown to modulate the distribution of EEs (Hoepfner et al, Cell 2005).*

We agree with the Reviewer that the identification of the anterograde kinesin counterbalancing our dual EE retrograde motor system is an important point for the manuscript. Therefore, although we are aware of the role of KIF16B in the control of EE localization, we identified another anterograde kinesin, namely **KIF5B**, as a candidate for this function. Indeed, **KIF5B** was reported as a MT plus-end directed kinesin modulating EE motility and traffic (Braun et al., 2016, *Curr Biol* 26: R1292-R1294; Nath et al., 2007, *Mol Biol Cell* 18: 1839-1849; Loubéry et al., 2008, *Traffic* 9: 492-509). As described on **page 12** and shown in **Fig. S2E**, **we found KIF5B silencing to impair the EE peripheral distribution induced by simultaneous CilioD and AZ82 treatment**, suggesting its engagement when both EE retrograde transport motors, Dynein and KIFC1, are inhibited.

9. *Figure 1C: Why does the GFP-STIM CC mutant (deletion of the 3 coiled coil domains) show exactly the same electrophoretic motility than WT? In Fig. S1, delta CC2 and CC3 mutants migrate faster than WT.*

We agree with the Reviewer. We apologize, yet we swapped the blot in the previous version of Fig. 1C, now **Fig. 1E**, with the right one, showing also the **lower molecular weight gel part**.

10. *The authors claim that they verify "the ability of ER protein STIM1 to bind, similarly to p150Glued, EB1 (Grigoriev et al. 2008). The STIM1-p150Glued interaction has not been documented in that manuscript.*

We agree with the Reviewer. **We corrected the mistaken reference** to Honnappa et al. *Cell* 2009 (138: 366-376) which reports STIM1, p150Glued and EB1 as known MT plus end proteins.

Thank you for submitting a revised version of your manuscript. The study has now been seen by two of the original referees. Reviewer #1 was unfortunately not able to review the revised version. While both reviewers appreciate the added information, reviewer #2 also indicates issues with the newly included biochemistry experiments that would have to be clarified in the final revised manuscript. Therefore, I would like to invite you to address the remaining referee comments and the following editorial issues:

- 1) Please submit up to five keywords.
- 2) Please fill in and submit the Author Checklist: ([https://wol-prod-cdn.literatumonline.com/pb-assets/embo-site/Author Checklist %20-%20EMBO%20J-1561436015657.xlsx](https://wol-prod-cdn.literatumonline.com/pb-assets/embo-site/Author%20Checklist%20-%20EMBO%20J-1561436015657.xlsx)). The completed author checklist will also be part of the Review Process File.
- 3) We require a Data Availability Section at the end of Materials and Methods. As far as I can see, no data deposition in external databases is needed for this paper. If I am correct, then please state in this section: This study includes no data deposited in external repositories. Further information can be found at <https://www.embopress.org/page/journal/14602075/authorguide#data deposition>
- 4) Please rename supplementary figures into Figure EV1, Figure EV2 etc and update the figure legends and callouts in the text. For further details, please see <https://www.embopress.org/page/journal/14602075/authorguide#expandedview>
- 5) Please rename movie files into "Movie EV1" etc. and update their legends and callouts accordingly. Please remove the movie legends from the manuscript text and zip them with each respective movie file. For further details, please see <https://www.embopress.org/page/journal/14602075/authorguide#expandedview>
- 6) Thank you for submitting source data for the figures. Please arrange source data in one file per figure, e.g. by zipping together the pdf and excel files belonging to one figure.
- 7) We noticed some discrepancies between the Western blot panels in the main figures and the source data, specifically for Fig1A, p150 Glued blot (IP) and for the STIM1 panel in Fig S3A. Please check that the provided images are correct.
- 8) Fig 1E/F labels appear to be swapped in the source data - please check.

When preparing your letter of response to the referees' comments, please bear in mind that this will form part of the Review Process File, and will therefore be available online to the community. For more details on our Transparent Editorial Process, please visit our website: http://emboj.embopress.org/about#Transparent_Process

Please feel free to contact me if have any further questions regarding the revision. Thank you again for giving us the chance to consider your manuscript for The EMBO Journal. I look forward to receiving the final revised version.

Referee #2:

A role of STIM1 in motor delivery to endosomes is of interest. A role for protrudin and ORP1L in this process has been claimed and this could be another protein. While the authors performed a serious set of experiments, the biochemistry is all but convincing at many points. I provide a number of examples.

1. Fig 1A,B. p150 can easily be detected in the control lab while the quantification suggests that there is no signal at all.
2. Fig 5B. The quantification of the siKIFC1+Stim1 lane does not correspond to the WB (the signal for KIFC1 cannot be close to 0 because easily detectable). I guess that this is a problem in the quantification by chemoluminescence (and then? this is not explained in the Materials).
3. Fig 7 siHOOK1 not convincing silencing. Als in Fig 7A WB 2nd lane. Why does HOOK1 runs higher than in the control lane 1?

The content is interesting but the biochemical support is not optimal.

Referee #3:

The authors have met my previous comments/criticisms in a very satisfactory way.

Referee #2:

1. Fig 1A,B. p150 can easily be detected in the control lab while the quantification suggests that there is no signal at all.

Since there is no plot in Fig. 1B, the referee is likely referring to the Western blot (WB) and its quantification in Fig. 1A. The representative shown WB and the quantification cannot be exactly identical, as **the WB is representative of one experiment**, whereas **the plot shows the average** between the three independent biological replicates performed. **We quantified again** the WB lanes from our three biological replicates and generated a corresponding new plot in **Fig. 1A**. The **difference** between the new and the former quantifications and plot is **minimal, statistically not significant, and does not affect the conclusions**.

2. Fig 5B. The quantification of the siKIFC1+Stim1 lane does not correspond to the WB (the signal for KIFC1 cannot be close to 0 because easily detectable). I guess that this is a problem in the quantification by chemoluminescence (and then? this is not explained in the Materials).

The representative shown WB and the quantification cannot be exactly identical, as the **WB is representative of one experiment**, whereas **the plot shows the average** between the three independent biological replicates performed. **We quantified again** the WB lanes from our three biological replicates and generated a corresponding new plot in **Fig. 5B**. The **difference** between the new and the former quantifications and plot is **minimal, statistically not significant, and does not affect the conclusions**.

3. Fig 7 siHOOK1 not convincing silencing. Als in Fig 7A WB 2nd lane. Why does HOOK1 runs higher than in the control lane 1?

We improved the quality of the HOOK1 WB in **Fig. 7C**.

We mounted again the HOOK1 panel of Fig. 7A that in the previous version was slightly counterclockwise tilted.

Editor accepted the manuscript.

Corresponding Author Name:

Journal Submitted to:

Manuscript Number: